# High-coverage allele-resolved single-cell DNA methylation profiling reveals cell lineage, X-inactivation state, and replication dynamics

DNA methylation patterns at crucial short sequence features, such as enhancers and promoters, may convey key information about cell lineage and state. The need for high-resolution single-cell DNA methylation profiling has therefore become increasingly apparent. Existing single-cell whole-genome bisulfite sequencing (scWGBS) studies have both methodological and analytical shortcomings. Inefficient library generation and low CpG coverage mostly preclude direct cell-to-cell comparisons and necessitate the use of cluster-based analyses, imputation of methylation states, or averaging of DNA methylation measurements across large genomic bins. Such summarization methods obscure the interpretation of methylation states at individual regulatory elements and limit our ability to discern important cell-to-cell differences. We report an improved scWGBS method, single-cell Deep and Efficient Epigenomic Profiling of methyl-C (scDEEP-mC), which offers efficient generation of high-coverage libraries. scDEEP-mC allows for cell type identification, genome-wide profiling of hemi-methylation, and allele-resolved analysis of X-inactivation epigenetics in single cells. Furthermore, we combine methylation and copy-number data from scDEEP-mC to identify single, actively replicating cells and profile DNA methylation maintenance dynamics during and after DNA replication. These analyses unlock further avenues for exploring DNA methylation regulation and dynamics and illustrate the power of high-complexity, highly efficient scWGBS library construction as facilitated by scDEEP-mC.

As our understanding of the role of DNA methylation in development, health, and disease has grown[1,2], the necessity of understanding cell-to-cell differences in the distribution of this important epigenetic mark has become clear. Several techniques for profiling DNA methylation in single cells have been developed, many of which focus on increasing cellular throughput[3–7]. However, these methods provide very low CpG coverage per cell, which substantially limits our ability to draw conclusions about individual cells and makes direct cell-to-cell comparisons difficult. Additionally, low CpG coverage necessitates the summarization of methylation measurements over large genomic regions[8], resulting in cell groupings driven by large (often megabase-scale) DNA methylation features, such as replication-associated hypomethylation at partially methylated domains (PMDs)[9,10]. While this approach can distinguish broad cell clusters, the lack of coverage at short regulatory elements generally does not allow for more precise cell type identification or in-depth analysis of single regulatory

✉e-mail: hui.shen@vai.org; peter.laird@vai.org

elements. A single-cell whole-genome bisulfite sequencing method that balances CpG coverage and cellular throughput would allow improved insight into diverse populations of cells while facilitating direct cell-to-cell comparisons. Such a method could profile small but mechanistically relevant features such as enhancers and promoters in individual cells, allowing interpretation of biologically functional methylation features on a per-cell basis.

The limitations of current scWGBS methods and analyses also restrict our insight into regulatory processes involving DNA methylation. During DNA replication, newly incorporated cytosines are unmethylated, and maintenance methylation is necessary to restore symmetric methylation at these hemi-methylated sites. Currently, our knowledge of this maintenance process is based on studies of bulk populations, often leveraging perturbations such as BrdU or EdU treatment[11,12]. While perturbation-free single-cell replication-sequencing methods have been developed[13,14], they provide no information on DNA methylation. A high coverage scWGBS method, which also provided accurate copy number calls across the genome, could provide insights into the dynamics of DNA methylation dynamics and maintenance in single replicating cells, without requiring perturbations such as BrdU.

Existing scWGBS methods and analyses typically do not provide allele-resolved methylation (ARM) calls, due to data sparsity and the ambiguities introduced by bisulfite conversion. ARM analyses could permit analysis of features such as imprinting and X-inactivation and allow analysis of hemi-methylation (asymmetric methylation on the reference and complement strands of the same allele at a given CpG). Read-backed phasing offers a promising approach for assigning individual reads to local haplotypes based on single-nucleotide polymorphisms (SNPs), but current tools cannot account for bisulfite-conversion induced sequence changes[15]. Conversely, bisulfite-aware approaches such as SNPSplit[16] require creation of custom reference genomes, time-consuming re-alignment, and a database of phased SNPs, which is often not readily available.

Here, we present an improved scWGBS method, which we term scDEEP-mC (single-cell deep and efficient epigenomic profiling of methyl-C). scDEEP-mC libraries display consistent, complete bisulfite conversion and have high complexity, allowing high CpG coverage at moderate sequencing depths. Additionally, we describe analyses designed to demonstrate the deep biological insight that can be obtained from this rich dataset. We demonstrate how this high-coverage scWGBS data can be leveraged to provide interpretable, granular cell type calls. We describe an improved algorithm for rapid and bisulfite-aware ARM calling in single cells. We query allele-specific methylation and population-specific hemimethylation enrichment and profile DNA methylation dynamics in single actively replicating cells. Finally, we identify X-inactivation state in single cells, even in the absence of phased SNP information.

## Results

### Library generation and technical performance

scDEEP-mC is optimized to provide high coverage at moderate sequencing depth by efficient production of complex libraries (Fig. 1a). scDEEP-mC is based on the post-bisulfite adapter tagging (PBAT) approach[17], which typically incorporates a cleanup step after bisulfite conversion to remove high-concentration NaHSO₃. However, this results in DNA loss, especially with low input DNA quantities. To prevent such loss, we sorted cells directly into a small volume of high-concentration sodium-bisulfite-based cytosine conversion buffer. After performing bisulfite conversion, we diluted the reaction until the concentration of NaHSO₃ was low enough to allow polymerase activity. The first strand was then synthesized by seven rounds of random priming with tagged random nonamers. We designed the base composition of the tagged random nonamer to complement that of the bisulfite-converted genome (49% A, 20% C, 30% T, and 1% G exclusively in CpG

context). We carefully titrated the primer concentration to minimize off-target priming events that result in adapter dimers and concatemers.

After exonuclease digestion of single-stranded fragments and a solid phase reverse immobilization (SPRI) cleanup to remove small fragments, second-strand synthesis was conducted via random priming with tagged nonamers (Fig. 1a). The base composition of these tagged random nonamers was adjusted to complement the predicted composition of the synthesized first strand (30% A, 20% G, 49% T, plus 1% C exclusively in CpG context). We note that the base composition adjustments described above minimize off-target priming and permit the construction of directional libraries, which allows for more efficient alignment. This stands in contrast to most random-priming-based approaches, in which priming events are randomly distributed across all available fragments, including other primers and both DNA strands, resulting in significant primer dimer contamination and non-directional libraries. Small fragments were removed with a second SPRI cleanup, and the double-stranded, tagged molecules were amplified using indexing PCR (Fig. 1a).

We performed scDEEP-mC on primary cells isolated from mouse intestinal epithelium and cultured primary human fibroblasts and evaluated its performance in comparison to publicly available scWGBS datasets, focusing on those with the highest reported CpG coverage[3–7,18]. To avoid biases introduced by widely varying analysis methodologies, we processed all raw sequencing datasets through a standardized analytical pipeline based on BISCUIT[19]. We first evaluated bisulfite conversion by measuring cytosine conversion rates and found that scDEEP-mC, as well as most other bisulfite-based methods, display reliably high cytosine conversion rates in the CpY context. Some variability is seen in CpA conversion, which may be due to biological methylation in this context in certain cell types. Notably, the PBAL and Cabernet methods display poorer CpY conversion rates, which may be attributable to the bead-based bisulfite conversion and enzymatic cytosine conversion methods utilized in these protocols, respectively (Fig. 1b, e).

Next, we compared the sequencing efficiency and coverage of scDEEP-mC libraries and those generated by other scWGBS methods. Hyperdiploid and cancer cells were excluded from analysis, as the larger amounts of DNA in these cells can artificially inflate library complexity compared to diploid cells. We also excluded suspected doublet libraries with high methylation discordance (see "Methods"). We evaluated sequencing efficiency by tracing a subset of reads through the analysis pipeline and measuring the fraction of bases that were retained or discarded in each step. This approach includes metrics such as alignment rate and duplicate rate, but also measures other sources of inefficiency such as adapter contamination, poor bisulfite conversion, and overlap between read pairs. Since some metrics, such as duplicate rate, are highly correlated with sequencing depth, two million reads were randomly sampled from each library for this analysis, allowing comparisons between deeply and shallowly sequenced libraries.

We found that scDEEP-mC displays minimal adapter contamination and very high alignment rates (especially compared to other PBAT-based methods, such as scBS-seq, scM&T-seq, scTrio-seq, and PBAL). Additionally, our random primer design reduces GC content bias compared to other random-priming-based approaches, permitting more even coverage of the genome (Supplementary Figs. 1 and 2). Overall, scDEEP-mC displays the highest sequencing efficiency of the methods studied (Fig. 1c, f). We also measured the genomic coverage of each library. scDEEP-mC libraries have high yield, allowing sequencing deep enough to cover 30% of CpGs at moderate sequencing depths (20 M reads/cell), even in primary cells and with very strict read-level quality filtering (Fig. 1d, g). While snmC-seq2[6] produces libraries with high sequencing efficiency (comparable to scDEEP-mC), its very low library yield severely limits sequencing depth and coverage. An alternative tagmentation and enzymatic conversion-based method, Cabernet[18], achieves genomic coverage comparable to scDEEP-mC, but suffers from adapter contamination and incomplete

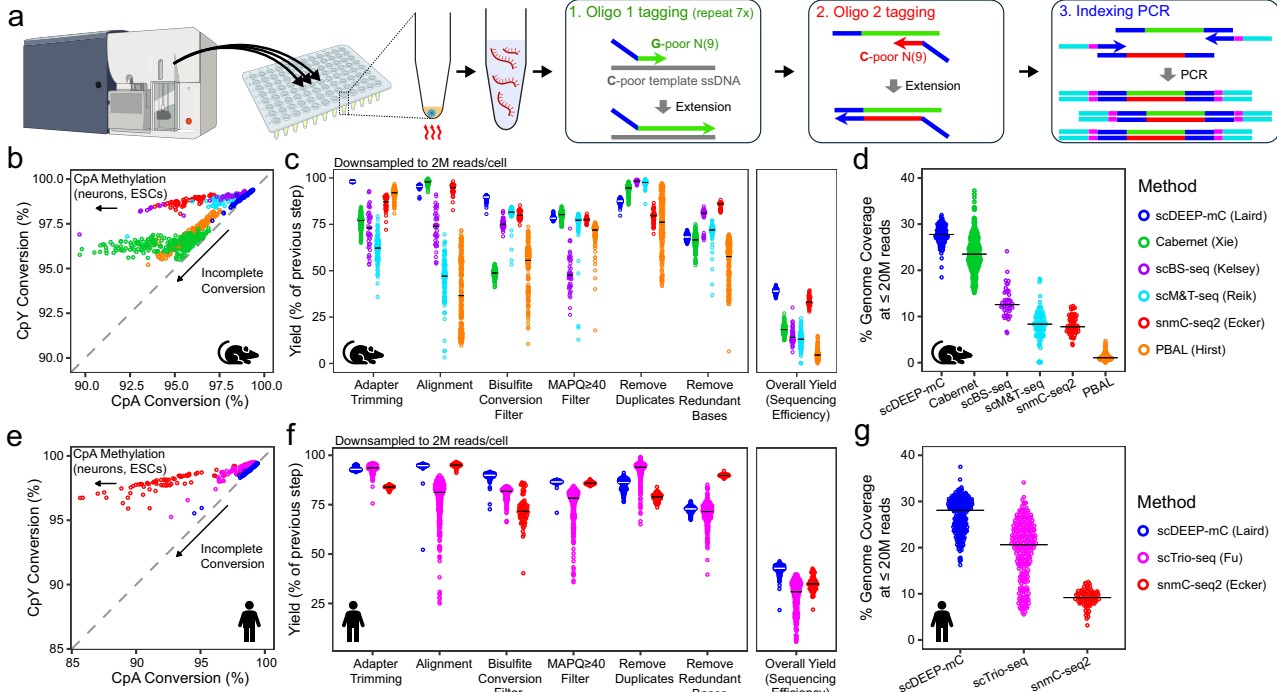

**Fig. 1 | scDEEP-mC provides high coverage in single cells by efficient production of high-complexity libraries. a** Library preparation overview. Single cells derived from culture or primary tissue are flow-sorted into bisulfite conversion buffer in a 96-well plate. Heating facilitates cell lysis, bisulfite conversion, and DNA fragmentation in one step. A 20× dilution step replaces purification after bisulfite conversion, eliminating DNA loss. First-strand adapters are added via seven rounds of base-composition-matched random priming, followed by second strand adapter tagging (also via random priming) and indexing PCR. **b–g** Comparison of scDEEP-mC with other scWGBS protocols. Publicly available raw sequence data from mouse (**b–d**) and human (**e–g**) cells were downloaded from[3–7,18] and analyzed using a standardized, Biscuit-based[19] pipeline (see "Methods"). Potential doublets and hyperdiploid cells were excluded. **b, e** scDEEP-mC displays reliable, complete bisulfite conversion, as evidenced by consistently high CpY conversion. **c, f** Two million reads were randomly sampled from each cell and the number of bases remaining after each processing step was recorded. From left to right, processing steps include removal of sequencing adapters, mapping, removal of reads with ≥1 CpY retention event, removal of reads with MAPQ < 40, and removal of duplicate reads. "Remove redundant bases" denotes the proportion of bases that contribute to coverage, and "overall yield" represents the number of covered bases per sequenced base. The median of each distribution is noted with a crossbar. scDEEP-mC displays very low adapter contamination and high alignment rates (especially compared to scBS-seq, to which it is most similar). Overall, scDEEP-mC achieves the highest efficiency of all compared methods. **d, g** Genomic coverage attained at a sequencing depth ≤20 M reads in all compared cells, with strict quality filtering. Cell numbers: **b** 774 cells, **c** 749 cells, **d** 793 cells, **e** 551 cells, **f** 542 cells, **g** 552 cells. **a** includes graphics created in BioRender. Foy, K. (2025) https://BioRender.com/a91w842. Source data are provided as a Source Data file.

cytosine conversion, which can bias methylation measurements. We found that Cabernet libraries display incomplete cytosine conversion at CpY in 43% of reads in human libraries, and 49% in mouse libraries. As a result, the Cabernet method displays inefficient sequencing yield after strict bisulfite conversion filtering to remove reads with evidence of CpY retention. However, even strict filtering does not eliminate the methylation bias resulting from incomplete conversion of unmethylated CpGs in the remaining reads. Overall, scDEEP-mC represents an excellent combination of consistent bisulfite conversion, high library complexity, and high CpG coverage.

## Cell type and state analysis

Identification of cell type and state is an important first step in many single-cell DNA methylation studies. To demonstrate the utility of high-coverage, high-efficiency scWGBS, we leveraged scDEEP-mC data to identify cell types in an unselected population of primary cells isolated from mouse intestinal epithelium. We demonstrate three approaches to this question, with varying levels of simplicity, interpretability, and supervision.

Analysis of cell-type-specific hypomethylated regions lifted over from human atlas data illustrates a straightforward classification approach. For each cell, we measured the mean beta for all CpGs in each of 50,286 cell-type-specifically hypomethylated regions cataloged by Loyfer and colleagues[20], then summarized by target cell type. In this analysis, cells corresponding closely to the target cell type will display significant hypomethylation. Most cells can be clearly assigned into epithelial or immune categories, while a few cells with intermediate methylation states in immune- and epithelial-specific regions represent doublets. We also observed two stromal cells with notable hypomethylation in muscle-specific regions. We further measured pairwise concordance (Hamming distance) between all pairs of cells and found high similarity within cell types (Supplementary Fig. 3).

To increase the resolution of these cell type calls, we summarized methylation levels at each gene promoter for each cell and used hierarchical clustering to identify groups of cells with similar promoter methylation phenotypes (Fig. 2b). This approach exactly recapitulated the cell assignments derived from cell-type-specific hypomethylation analysis but distinguished additional subgroups and facilitated direct analysis of epigenetic regulatory differences between clusters. Specifically, we identified two major groups of cells, enterocytes and immune cells (Fig. 2c), each containing two sub-groups. In one enterocyte subgroup, promoters of genes involved in absorptive processes (such as *Abcg5*) were unmethylated, while in the other subgroup, promoters of genes implicated in differentiation and development (*Nkx1-2* and *Satb2*) were unmethylated, suggesting that these two subtypes represent mature enterocytes and less differentiated cells, respectively (Fig. 2d). Within the immune group, we inferred that the two groups represented T cells (hypomethylated at promoters of genes including *Itk* and *Cd3* complex members) and B cells (hypomethylated at promoters of genes including *Blnk*) (Fig. 2e).

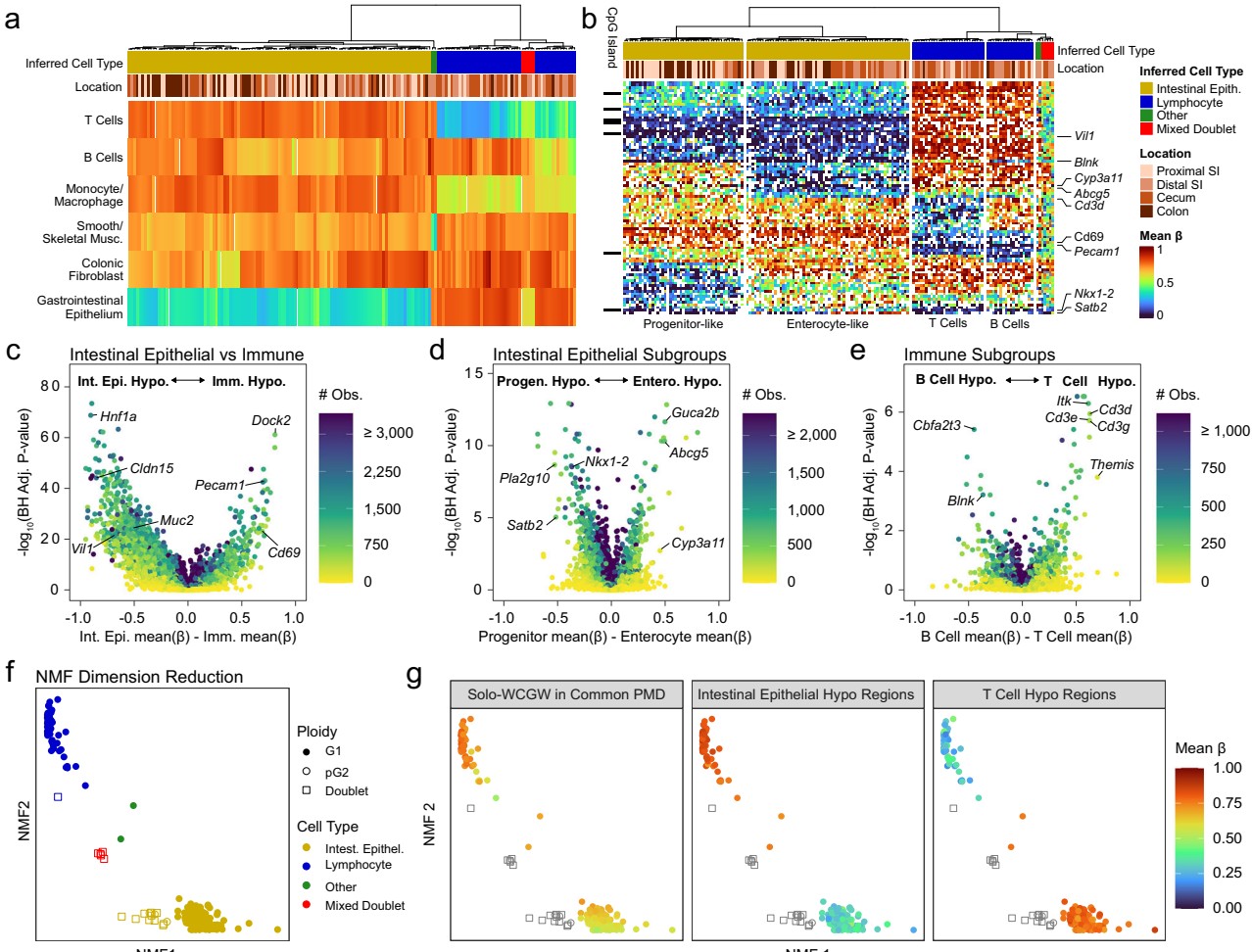

**Fig. 2 | scDEEP-mC captures high-resolution cell type and state information.** **a** Unsupervised hierarchical clustering of n = 164 high-coverage cells and doublets from primary mouse intestinal epithelium by mean methylation state in cell-type-specific hypomethylated regions[20]. Columns represent individual cells, while rows represent aggregated cell-type-specific region sets. Gastrointestinal epithelial and lymphocytic cell types can be clearly distinguished, as well as several doublets incorporating different cell types with intermediate beta values. **b** Unsupervised hierarchical clustering of differentially methylated promoters exactly recapitulates the cell types found in (**a**), but with additional granularity and biological insight. Most cell-type-specific promoters do not incorporate CpG islands. **c**–**e** Volcano plots illustrating differentially methylated promoters between cell groups (P values from unpaired two-sided t-tests, subject to Benjamini-Hochberg correction). The number of methylation calls contributing to each test is shown on the color scale. **f** Rank-2 NMF dimension reduction of raw beta values at 4.7 million variably methylated CpGs in n = 175 cells (including 11 low-coverage cells) perfectly recapitulates the cell types discovered in (**a**). Note that doublets and putative G2-phase cells (see Fig. 4c) are segregated from their respective cell types. **g** Cell-type-specific hypomethylation (as in [**a**]) and solo-WCGW methylation values overlaid on each cell in the NMF map. Source data are provided as a Source Data file.

Finally, we applied a completely unsupervised dimension reduction technique to this dataset. While these techniques may pose interpretability challenges, they can be appealing due to their unbiased nature. We chose non-negative matrix factorization (NMF) for this task due to its ability to natively accommodate sparse data. We applied rank-2 NMF to raw beta values of the 75% most variable CpGs across all cells, essentially decomposing this high-dimensional methylation data into two dimensions. This analysis revealed two major cell clusters (lymphocytes and intestinal epithelial cells), exactly recapitulated the cell type groupings previously described, and was also able to partition doublets and putative G2-phase cells (Fig. 2f). Overlaying the mean beta value in regions specifically hypomethylated in intestinal epithelial cells or T cells further confirms the accuracy of cell type inference by NMF (Fig. 2g).

### Allele-resolved methylation analysis
scDEEP-mC is especially well suited to investigate complex and poorly characterized DNA methylation phenomena such as hemi-methylation (asymmetric methylation on the reference and complement strands of the same allele). In general, three data points are necessary to accurately measure hemi-methylation: the cell, allele, and strand from which a methylation call originates. Strand assignment is trivially performed during alignment, but determining cell and allele simultaneously has proved difficult. Hairpin-based approaches constructed from bulk DNA samples allow for confident assignment of methylation calls to the same cell and allele but cannot correlate information across multiple reads to generate single-cell profiles[21]. Low-coverage scWGBS methods provide cell and strand information, but the small number of CpGs covered by these methods limits the utility of allele-resolved analyses. High-coverage, high-efficiency scWGBS (as embodied by scDEEP-mC) thus represents an excellent opportunity to investigate hemimethylation directly by extracting ARM calls, without relying on cellular perturbations[22], specialized library construction techniques[21,23], or indirect inference[18].

We developed a bisulfite-aware, fully reproducible Nextflow[24] pipeline (see "Methods" and Supplementary Information) to assign methylation calls to alleles by analysis of within-read heterozygous SNPs. The pipeline supports discovery of suitable SNPs directly

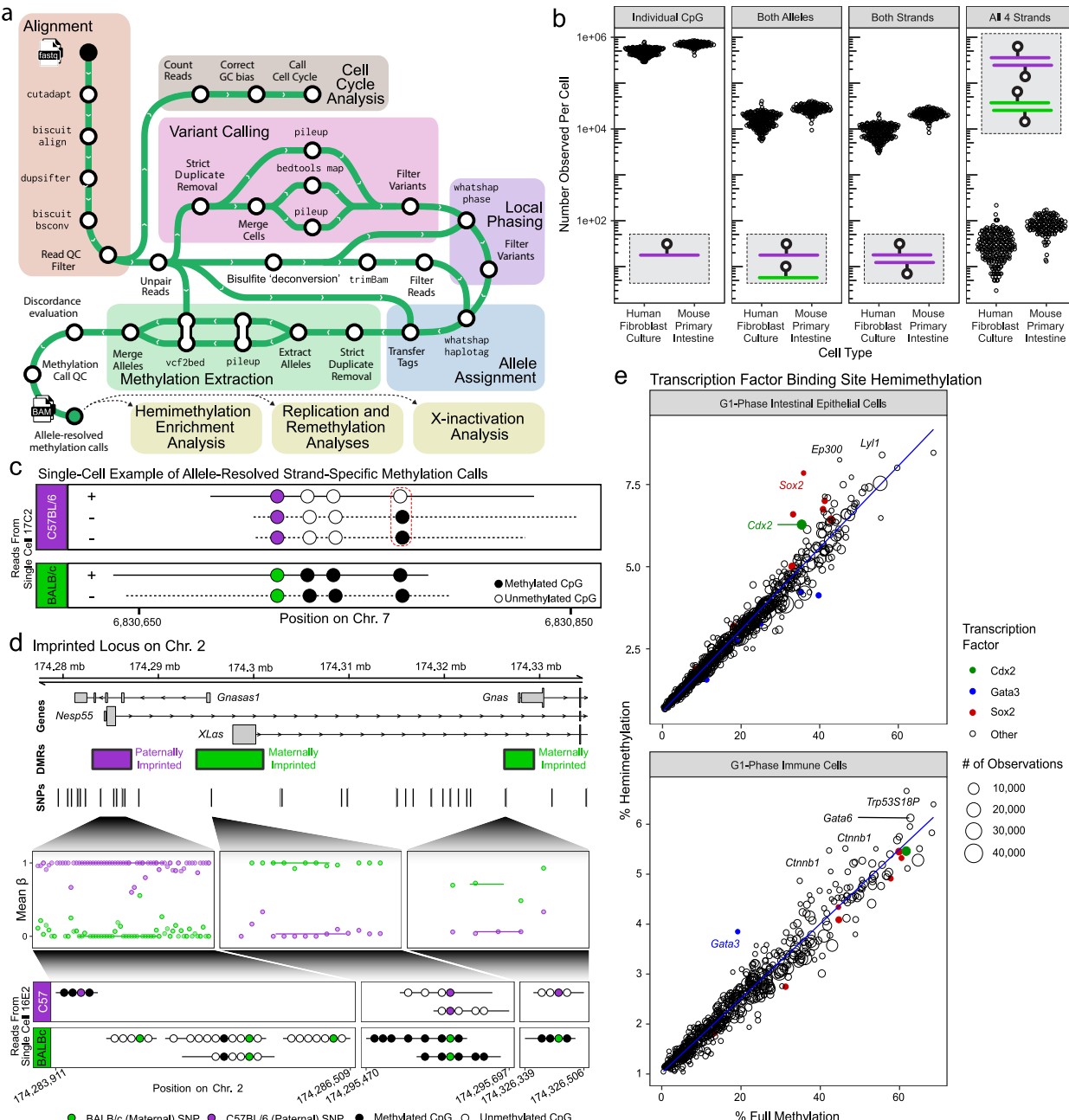

**Fig. 3 | scDEEP-mC facilitates high-resolution, genome-wide analysis of allele-resolved methylation, including hemi-methylation. a** Overview of allele-resolved methylation analysis. High-quality heterozygous SNPs are discovered and used to assign reads to local alleles (see "Methods"). **b** Allele-resolved CpG coverage. In primary mouse cells (known SNPs) and human cells (SNPs discovered de novo), ~1 million CpGs are assigned to an allele. Approximately 10,000 loci per cell have two-strand coverage, allowing for hemi-methylation measurement. **c** Example locus showing reads from a single cell providing information from all four strands (reference and complement, each allele), highlighting hemi-methylation and allele-specific methylation. **d** Example imprinted (Gnas) locus. Reads from one cell define all three differentially methylated regions. **e** Cell-type-specific enrichment of hemi-methylation in TFBS. Hemi-methylation content is plotted against (symmetric) methylation frequency, since the absolute quantity of hemi-methylation is strongly correlated to TFBS methylation level. Source data are provided as a Source Data file.

from the input data; alternatively, a database of high-quality known SNPs can be provided. In contrast to existing algorithms[16], this approach does not require creation of a custom reference or time-consuming re-alignment (Fig. 3a). Analysis of SNPs discovered by our pipeline yielded a precision of 93.5% and recall of 67.4% compared to high-quality SNPs reported by the Mouse Genome Project, and a precision of 90.8% and recall of 69.5% compared to four littermates (Supplementary Fig. 4a).

We tested this ARM analysis on primary intestinal epithelial cells from F1 offspring of a cross between two inbred mouse strains (C57BL/

6 and BALB/c), as well as cultured primary human fibroblasts. In both groups, we generated ARM calls for nearly 1 million CpGs per cell, including ~10,000 CpGs with both strands covered (permitting analysis of hemi-methylation) per cell (Fig. 3b, c). Analysis of synthetic reads generated from an artificial heterozygous genome revealed that our pipeline had a read-assignment precision of 99.97% and recall of 76.8% (Supplementary Fig. 4b). ARM analyses highlight the high coverage of scDEEP-mC; for example, a single cell is sufficient to define all three differentially methylated regions on both alleles of the imprinted *Gnas* locus (Fig. 3d).

Since scDEEP-mC measures allele- and strand-resolved methylation states in single cells, it enables *post hoc* population-specific analyses of hemi-methylation (Fig. 3e). We measured overall methylation and hemi-methylation at transcription factor binding sites (TFBS) in each of the major cell groups described in Fig. 2b and found that *Cdx2* binding sites were hemi-methylated more often than expected in intestinal epithelial cells, while *Gata3* binding sites were hemi-methylated more often than expected in immune cells (Supplementary Fig. 5). Additionally, *Sox2* binding sites had relatively high rates of hemi-methylation in intestinal epithelial cells. This finding is particularly interesting considering that high-throughput screening of transcription factors specifically identified SOX2 as a super pioneer factor that achieves demethylation by the inhibition of maintenance remethylation of hemi-methylated DNA[25]. This mechanism of demethylation by passive dilution of DNA methylation through repeated rounds of cell division stands in contrast to the active demethylation by TET enzymes, which seems to be the main mechanism utilized by most other pioneer factors[25].

## DNA methylation maintenance in actively replicating cells

Previous work has shown that CpGs in late-replicating loci are prone to DNA methylation loss over successive mitotic divisions[9,10], likely caused by incomplete remethylation of hemi-methylated CpGs produced by DNA replication[12]. To investigate this, we performed scDEEP-mC on both early-passage and very late-passage, TERT-immortalized human fibroblasts. Using publicly available replication timing data[26], we divided the genome into thirteen partitions, from very early- to very late-replicating. We then measured the relative amounts of unmethylation, hemi-methylation, and full methylation in each of these partitions (Fig. 4a). In early passage cells, late-replicating loci have higher proportions of hemi-methylation, consistent with the hypothesis of incomplete remethylation at late-replicating regions, as depicted in Fig. 4b. This incomplete maintenance of methylation leads to substantial loss of methylation at late-replicating regions after many mitoses in the late-passage cells. Both early- and late-passage cells display a proportional enrichment of hemi-methylation in late replicating regions (Fig. 4a, bottom panels), indicating persistent incomplete remethylation at these loci.

To further investigate the relationship between replication and DNA methylation, we sought to identify actively replicating cells. We reasoned that cells in S-phase would have higher copy numbers in (and thus more reads aligned to) early-replicating regions of the genome (Fig. 4c), as well as higher ARM discordance (frequency of conflicting methylation calls for the same cytosine) due to incomplete remethylation. We measured the fraction of reads aligning to early-replicating regions and ARM discordance for each fibroblast cell and were able to distinguish several categories of cells (Fig. 4d). G1 phase cells displayed a relatively even distribution of reads across the genome as well as low ARM discordance. S-phase cells had more reads in early-replicating regions as well as higher ARM discordance. Finally, we identified putative G2-phase cells with higher ARM discordance, but an even distribution of reads across replication timing bins. We also identified doublets with very high ARM discordance (>3%), which corresponded with the ARM discordance measured in high-confidence, mixed cell-type doublets in mouse cells.

We then quantified methylation state distribution and normalized read count in 50-kb bins across the genome in cells at various stages of replication (Fig. 4e). Cells in early S-phase have only a few regions with 4n copy number, but these regions display high levels of intermediate methylation due to incomplete remethylation of the daughter strand. As cells progress through S-phase, more regions of the genome acquire a 4n copy number, but the proportion of CpGs with intermediate beta values in these regions decreases due to maintenance re-methylation. Once the cell enters G2 phase, all regions have a 4n copy number, and the proportion of CpGs with intermediate beta values is still lower, but not diminished to the level seen in G1 cells (Fig. 4e).

## Whole-chromosome X-inactivation epigenetics in single cells

Existing methods for studying X-inactivation in single cells rely on transcriptomics[27], which provides very limited information from the inactive chromosome (zero or a few transcripts per cell from regions such as *Xist*). In contrast, when paired with allele-resolved analysis, scDEEP-mC can characterize the chromosome-wide DNA methylation state of both X alleles in single female cells. This includes the status of hundreds of promoters on both alleles, allowing us to call X-inactivation state with high confidence in each cell and offering deep insights into potential dysregulation of X-inactivation. As a proof of concept, we applied our ARM analysis to scDEEP-mC-sequenced cells from a C57BL/6 x BALB/c F1 mouse, assigning reads from chromosome X to the maternal or paternal allele utilizing high-quality phased heterozygous SNPs from the Mouse Genome Project.

To determine which X chromosome was active in each cell, we measured the mean beta value across all CpG islands on each allele for each cell (Fig. 5a); notably, some (but not all) doublets had intermediate allele-specific CpG island methylation states, representing a pair of cells with opposite X-inactivation states. Based on the X-inactivation calls defined by this analysis, we then measured the mean beta value of each CpG island and promoter on the inferred active and inactive X chromosomes across all cells (Fig. 5b, c). As expected, we observed that CpG islands (CGIs) and promoters were generally methylated on the inactive X chromosome (Xi) and unmethylated on the active X chromosome (Xa). Some CGIs were methylated on both alleles, while only the CGIs proximal to the Xist and Dxz4 macrosatellite loci[28] were methylated on Xa and unmethylated on Xi (Fig. 5b). Two SNP-proximal regions were identified that were unmethylated on both alleles. *Nap1l3* is known to escape X-inactivation[29], while the *Pcdh11* CpG island is located in a gene body. The promoters of genes on Xi exhibited higher methylation levels than those on Xa, while gene bodies displayed slightly higher methylation levels on Xa (Fig. 5c).

Since the behavior of CGI methylation across Xi is highly correlated, we investigated whether we could infer X-inactivation status analytically, even in the absence of chromosome-wide SNP phasing. If successful, this approach would permit the study of X-inactivation in any heterogeneous population of female single cells, obviating the need for long-read sequencing and phasing of SNPs. We piloted this analysis in the mouse cell dataset, where the availability of high-quality phased SNPs from the Mouse Genome Project allowed us to validate our analysis quantitatively (Fig. 5d). We chose to use NMF for this analysis, since it allows us to specify the number of expected components and because of its ability to generalize over sparse input data. We defined a feature for each allele-resolved CpG on chromosome X (Fig. 5e), then used rank-2 NMF to factorize our dataset. This process, which essentially clusters cells by allele-resolved chromosome X methylation state (Fig. 5f), yielded two factor loadings for each cell and allele-resolved CpG. We found that the cells separated into two groups, each enriched for a different NMF factor, and representing opposite X-inactivation states when analyzed using known, phased SNPs (Fig. 5g). Likewise, allele-resolved CpGs in or near CpG islands separated into two groups representing the ground-truth alleles determined by analysis of phased SNPs (Fig. 5h).

We then assigned cells and loci to NMF-derived cell groups (representing two distinct X-inactivation states) and NMF-derived alleles and computed the mean beta value for each feature across all cells in each group. We found that the methylation status measured by NMF-inferred cell groups and alleles correlated very well with ground truth measurements ($R^2 = 0.99$) (Fig. 5i). Comparing the distribution of methylation states across ground truth (Fig. 5j) and inferred (Fig. 5k) allele assignments for all cells also reveals high concordance between these approaches. This advance enables the analysis of X inactivation states in any heterogeneous population of single female cells directly from short-read bisulfite sequencing data, even when phased SNPs are not available.

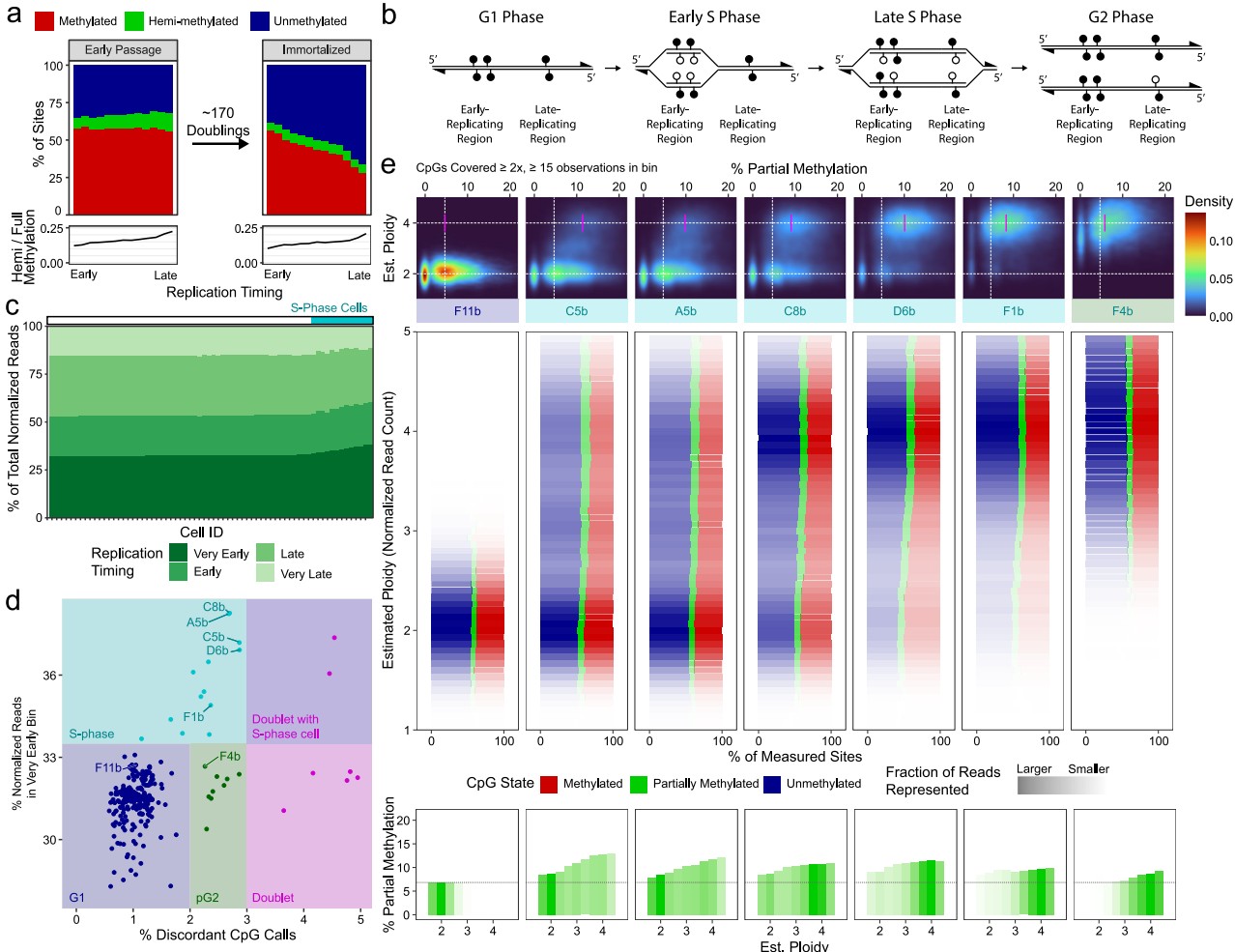

**Fig. 4 | scDEEP-mC enables profiling of replication-induced partial methylation and re-methylation in actively replicating cells. a** Quantification of two-strand methylation state across the replication timing spectrum in human fibroblasts. In early-passage cells, more hemi-methylation is present in late-replicating regions than in early-replicating regions. Over many mitoses, incomplete maintenance of methylation at these loci leads to loss of methylation. Notably, the amount of hemi-methylation relative to symmetric methylation remains relatively unchanged. **b** Diagram illustrating methylation dynamics during replication. Semi-conservative replication of 5-methylcytosine results in hemimethylation, which is slowly resolved by maintenance methylation. **c** Identification of replicating cells. The fraction of normalized reads (see "Methods") assigned to very early, early, late, and very-late-replicating regions of the genome is shown for a selection of cells, sorted by the fraction of reads in the "very early" bin. S-phase cells have more reads in earlier-replicating bins. **d** The fraction of reads in very-early-replicating regions

(Fig. 4b) is plotted against the fraction of discordant methylation calls for each cell. Cells in S-phase have more reads in earlier-replicating regions of the genome and higher discordance; putative G2-phase cells (pG2) exhibit higher discordance but even read distribution. Doublets exhibit very high discordance. **e** Distribution of methylation states in replicating cells. Seven single cells in various stages of replication are shown, ordered by their replication progress. For each cell, the estimated ploidy and proportion of CpGs with partial methylation was calculated for 50 kb windows across the genome. The top panels show distribution of bins by estimated ploidy and fraction partial methylation; the modal % partial methylation for replicated regions is marked in magenta. Middle panels show the distribution of methylation states for each cell, aggregated by estimated ploidy; the distribution of reads is depicted by the opacity of the bar (regions with more reads are more opaque). Bottom: same data as middle panels, focusing on % partial methylation. Source data are provided as a Source Data file.

## X-inactivation dynamics in serially cultured human fibroblasts

Next, we examined the effect of extended culture on X inactivation in female human fibroblasts (Fig. 6a). When we measured the standard deviation of allele-resolved beta values in CGIs across cells, we noticed that early-passage cells exhibited high variance on chromosome X, reflecting the random assortment of X-inactivation states within the population. Late and (TERT-immortalized) very late passage cells, however, had diminished ARM variance on chromosome X, indicative of a loss of X-linked CGI methylation diversity between cells (Fig. 6b). Several mechanisms could explain this observation, including population-wide reactivation of Xi (accompanied by loss of CGI methylation), population-level chromosomal loss of Xi (with or without reduplication of Xa), and evolutionary drift in the population favoring one X-inactivation state (Fig. 6c). Notably, both loss of Xi (scenario 3) and evolutionary drift (scenario 4) result in a homogenous

population of cells expressing transcripts from one copy of one parent's X. Thus, in transcription-based single-cell X-inactivation studies, these scenarios could only be distinguished by searching for the presence of Xist transcripts, which (given the sparsity of single-cell transcriptomic data) could result in substantial ambiguity. In such scenarios, it could be difficult or impossible to confidently determine whether all cells have retained both X alleles using RNA data. scDEEP-mC, however, can provide allele-resolved calls on numerous sites across both alleles, allowing confident interpretation of X-inactivation status in each cell.

As described above, we used rank-2 NMF to identify two highly correlated groups of cells and loci on chromosome X in early-passage cells. We then used the NMF alleles determined from early-passage cells to analyze late- and very-late-passage cells. We observed similar numbers of CpG methylation calls in CpG islands and shores from each NMF

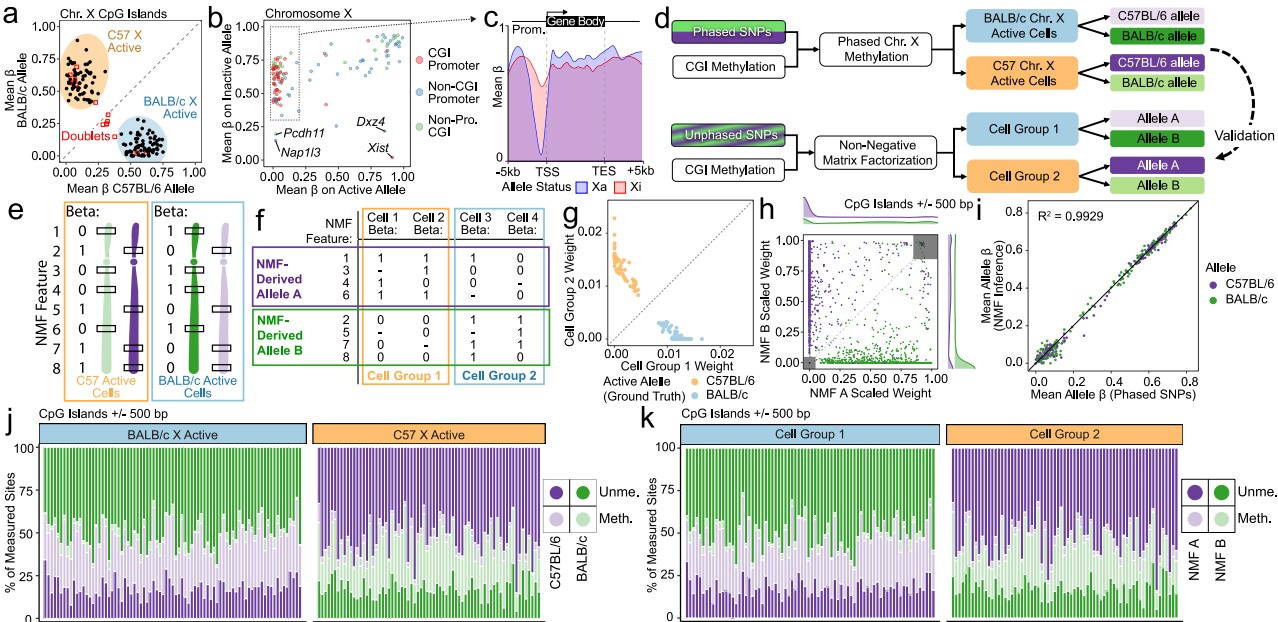

**Fig. 5 | scDEEP-mC enables quantification of whole-chromosome X-inactivation epigenetics in single cells, with or without phased SNPs. a** Mean CpG island methylation on the active and inactive X chromosomes in *n* = 151 cells from primary mouse intestine. **b** Mean methylation on the Xa and Xi (as defined in [**a**]) for promoters and CpG islands on chromosome X, across all single cells. **c** Metagene plot showing methylation levels across genes subject to X-inactivation (outlined in [**b**]). **d**–**f** NMF can be used to model X-inactivation in populations of cells by bipartitioning highly correlated methylation states across cells and chromosomes. High-confidence phased SNPs included in the Mouse Genome Project were used as the ground truth. **g**, **h** NMF analysis almost perfectly recapitulates ground-truth data, correctly assigning cell X-inactivation state (**g**) and allele membership (**h**). **i** Mean methylation values of loci on methylation-derived alleles computed by NMF correlate with the corresponding ground-truth values. **j**, **k** CpG island methylation distribution is shown for each cell; colors correspond to allele, while saturation denotes methylation state. Ground truth data from phased SNPs (**j**) or NMF-derived alleles (**k**) are shown. Source data are provided as a Source Data file.

allele across all passages, indicating that both NMF alleles are present in all passages (Fig. 6d). As expected, early-passage cells partitioned into two groups with opposite allele-resolved CpG island methylation states. However, in late-passage cells, most cells displayed methylation of CpG islands on NMF allele A, while all cells in the very late passage displayed methylation of CpG islands on NMF allele B (Fig. 6e). (We note that the very late passage cells share a common parent with the late passage cells but are not directly descended from them.) This illustrates that while X-inactivation regulation appears largely intact in the later-passage cells, a single X-inactivation state tends to become prevalent after many passages, most likely due to evolutionary drift.

Finally, we utilized single-cell read count data to perform copy number analysis. This also ruled out X-chromosome loss in any passage (Fig. 6f). However, this analysis did reveal two major aneuploid subclones with chr8 (Cluster 2) and chr10q (Cluster 3) deletions, respectively, in the very-late-passage cells. Previous work has shown that isolated CpGs in a WCGW context, known as solo-WCGWs, tend to lose methylation in a predictable manner that is correlated with cumulative cell divisions[9,10]. Notably, the clusters we identified in these very-late-passage cells exhibited differing levels of solo-WCGW methylation (Fig. 6g), reflecting differences in replicative history. They also display varying CGI methylation (Fig. 6h), illustrating the power of scDEEP-mC to characterize subtle but important differences in relatively homogenous cell populations.

## Discussion

As we have shown, scDEEP-mC represents an advance in single-cell whole-genome bisulfite sequencing methods. By combining high library complexity with high CpG coverage, scDEEP-mC facilitates detailed and interpretable analyses of DNA methylation features in single cells. Besides contributing additional resolution and granularity to common analyses such as cell classification, we also demonstrate how scDEEP-mC facilitates ARM analyses in single cells. Notably, the

SNPs required for distinguishing between alleles can be discovered de novo directly from scDEEP-mC sequencing data, simplifying experiment design and expanding the range of potential applications for this technology.

When coupled with allele-resolved analysis, scDEEP-mC enables the investigation of complex DNA methylation phenomena, such as hemi-methylation and X-inactivation, at the single-cell level. This also permits group-wise analyses of these phenomena in novel cell populations identified via *post hoc* analyses. For example, we demonstrate how scDEEP-mC can be used to characterize enrichment of hemi-methylation at TFBS in distinct cell populations.

Our results also highlight the potential of scDEEP-mC to investigate DNA methylation maintenance dynamics at the single-cell level. By simultaneously measuring replication status (via copy number analysis) and DNA methylation of individual genomic regions in single cells, scDEEP-mC allows unprecedented insight into genome-wide maintenance methylation dynamics. Similarly, scDEEP-mC (when paired with ARM analyses) allows detailed analysis of chromosome X epigenetics in single cells. Since scDEEP-mC generates epigenetic information across the entire inactive X chromosome, it permits higher-confidence, more complete analysis of the inactive X chromosome than transcription-based methods. As an illustration of this ability, we demonstrate how scDEEP-mC can distinguish between population-wide loss of the inactive X chromosome versus population drift toward a single X-inactivation state in serially cultured human fibroblasts.

The plate-based nature of scDEEP-mC limits the number of cells that can be analyzed in parallel, although miniaturization of the protocol has the potential to increase cellular throughput in the future. The deep sequencing performed on each cell also reduces the number of cells that can be analyzed for a given number of sequence reads. As with other random priming-based methods, scDEEP-mC is subject to some random priming bias (Fig. S2), although this effect has been

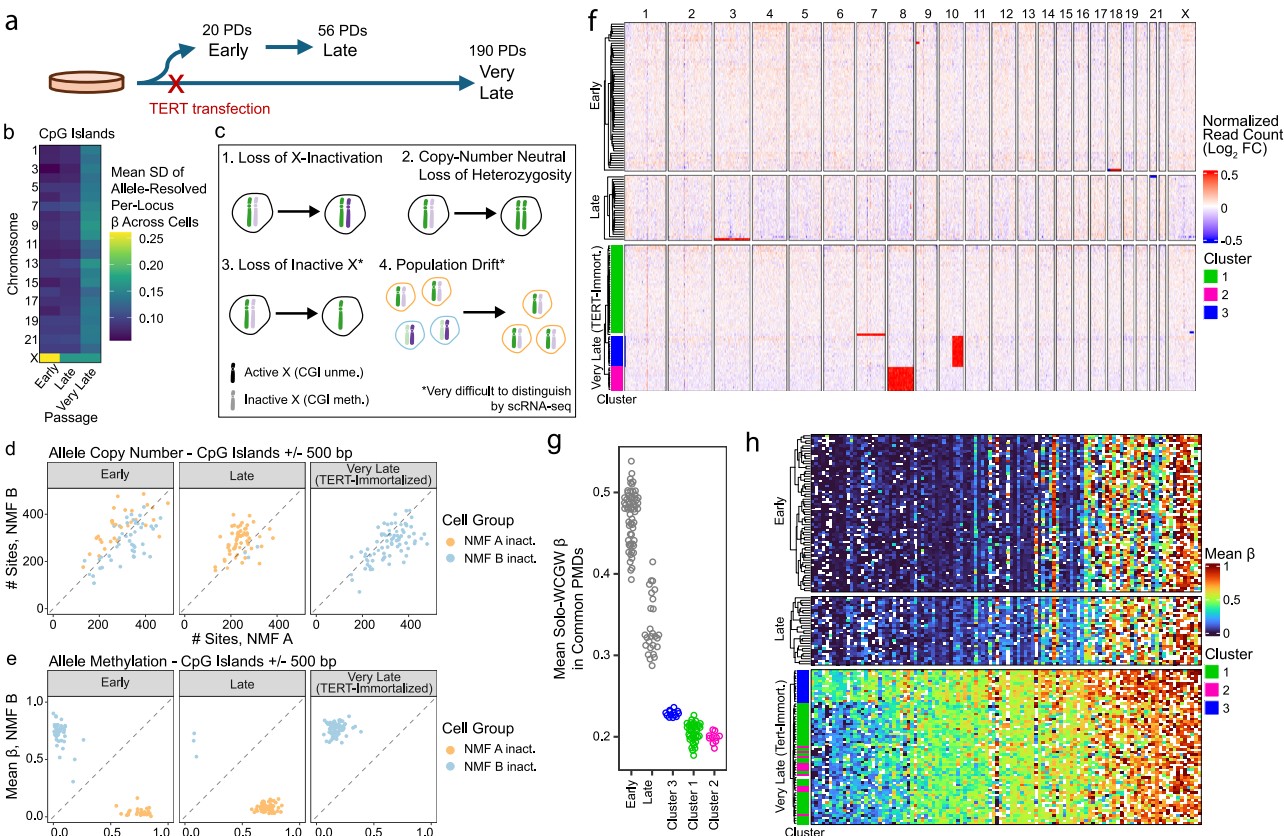

**Fig. 6 | scDEEP-mC illuminates X-inactivation changes and population heterogeneity in cultured human fibroblasts. a** Experimental overview. AG06561 cells were obtained from Coriell; a subculture was immortalized via TERT transfection and cultured for ~1 year. A parallel culture was passaged until senescent. **b** High between-cell variability of CGI methylation states on a single allele is noted on chromosome X in early passage cells, indicative of a mixture of X-inactivation states in the population. **c** Loss of variability in CGI methylation on the X chromosome in late and very late-passage populations suggests several possible explanations, some of which are very difficult to distinguish via expression-based methods. **d**, **e** NMF analysis of chromosome X methylation in early-passage cells groups loci into NMF-derived alleles. NMF alleles have similar numbers of methylation calls (**d**), indicating relatively stable copy number, but widely differing methylation states (**e**). Loss of variability in chromosome X CpG islands at late passage is due to population drift rather than copy number changes or dysregulation of X-inactivation. **f** Copy number analysis of early, late, and very-late passage cells confirms that no substantive copy number alterations are present on chromosome X and identifies three major subclones in very-late-passage cells (one diploid and two hypodiploid). These clusters have different mean solo-WCGW methylation values (**g**) as well as differential CpG island methylation state (**h**), illustrating the power of scDEEP-mC to uncover variation even within relatively homogenous cell populations. Source data are provided as a Source Data file.

partially mitigated by thoughtful design of the random primers. We note that the design of the random primers does not account for methylation of cytosines in CpH context. Although CpH methylation is rare in most cell types, it may be found at non-negligible levels in e.g. neuronal cells[8]. In this setting, our primer design could bias against regions with high CpH methylation (although CpH methylation is still accurately reported by scDEEP-mC, since only the first nine bases of each read is generated via random priming).

In conclusion, scDEEP-mC represents a powerful tool for single-cell DNA methylation studies, enabling high-coverage, efficient profiling of DNA methylation at the single-cell level. Its applications are diverse and far-reaching, from cell type identification to the study of complex epigenetic phenomena such as hemi-methylation and X-inactivation. We anticipate that scDEEP-mC will become a valuable resource for researchers seeking to understand the intricacies of epigenetic regulation in normal development and disease. Importantly, all analyses described here can be applied directly to primary cells and do not require cell culture or experimental interventions, thus unlocking studies of X inactivation and replication dynamics in single cells from primary human tissues. Future directions for this research could include applying scDEEP-mC to study DNA methylation maintenance in normal and diseased aging states, such as Hutchinson-Gilford

progeria[30]. Additionally, integrating scDEEP-mC with other single-cell omics approaches, such as STORM-seq[31], could provide a more comprehensive understanding of epigenetic regulation and the interplay between transcription and DNA methylation.

## Methods

All studies were conducted in accordance with relevant ethical regulations and were approved by the Van Andel Institute Animal Care and Use Committee (protocol #21-07-019).

### Primary cell culture and transduction

Primary human fibroblasts were obtained from the NIA Aging Cell Culture Repository at the Coriell Institute for Medical Research (Catalog ID AG06561) and maintained in Eagle's MEM with Earle's salts, non-essential amino acids (Gibco 11140-050), and 15% v/v fetal bovine serum. Cells were maintained at 37 °C, 5% $CO_2$, and 21% $O_2$. Low-passage primary fibroblasts were transduced with purified lentiviral particles containing expression vectors encoding human telomerase reverse transcriptase (TERT) as previously described[10].

Prior to flow sorting, cells were washed twice in PBS and incubated with 0.25% trypsin for 5 min, allowing the cells to dissociate from the dish. Growth media was then added to the cells, which were centrifuged

at 300 × g for 5 min. The resulting pellet was resuspended in flow buffer (HBSS without magnesium or calcium, with 5% FBS, 5 mM EDTA, and 1 µg/mL DAPI) with a volume sufficient to achieve 10⁶ cells/mL.

## Intestinal cell isolation

Tissue samples were collected from the small intestine, cecum, and colon of an 8-week-old CB6F1/J female mouse purchased from Jackson Labs (Strain # 100007). A female mouse was chosen for these studies to facilitate analysis of X-inactivation. For the small intestine and colon, the dissection involved collecting ~2 cm sections from the proximal and distal small intestine, and about 4 cm of the mid-colon. The tissue pieces were washed three times with cold PBS, then cut into 2 mm pieces and placed in 15 mL of 5 mM EDTA/PBS with 22.5 µL of 1 M DTT. The tissue was triturated, transferred to 15 mL of fresh 5 mM EDTA/PBS with 22.5 µL of 1 M DTT, and incubated for 15 min. The tissue was then washed with cold PBS twice to wash out EDTA. The tissue was resuspended in 10 mL collagenase IV solution (Stemcell Technologies 07909) with 100 µL DNase I (Ambion AM2235) and 10 µL ROCK inhibitor Y-27632 (Sigma-Aldrich Y0503), then incubated for 10 min at 37 °C. The intestine was then triturated four times with a total of 40 mL cold PBS, and the filtrate was collected by filtering the supernatant through 100 µm, 70 µm, and 40 µm filters. The supernatant was spun at 300 × g for 5 min at 4 °C. The pellet was then resuspended in 1 mL flow buffer (10 mL of PBS with magnesium and calcium, 200 µL FBS, 100 µL DNase I, and 10 µL Y-27632); cells were then counted using a Countess II instrument (Thermo Fisher). 10⁶ cells were transferred into 100 µL of flow buffer, and Fc receptors were blocked with 0.5 µg of TruStain FcX PLUS (anti-mouse CD16/32, BioLegend Cat. No. 156603) for 10 min on ice. Without washing the cells, 0.5 µg PE anti-mouse CD45.2 recombinant antibody (BioLegend Cat. No. 111103) and 1 µg Alexa Fluor 647 anti-mouse CD326 (EpCAM) antibody (BioLegend Cat. No. 118211) were added. Cells were incubated in the dark, on ice, for 30 min. Cells were then washed twice (2 mL flow buffer, centrifuge at 300 × g for 5 min at 4 °C). Finally, cells were resuspended in 500 µL flow buffer; DAPI (1 µg/mL) was added prior to flow sorting.

## Cell sorting

Flow sorting was performed using a BD FACSymphony™ S6 Cell Sorter. Live cells were selected for sorting based on scatter properties and DAPI negativity (Supplementary Fig. 6). BD FACSDiva™ Software and BD StepSort were used for data acquisition and 96-well sorting.

## scWGBS library preparation, quality control, and sequencing

Cells (n = 175 mouse intestinal, n = 292 human fibroblast) were FACS sorted into 3 µL of freshly prepared CT Conversion Reagent (Zymo EZ DNA Methylation-Gold Kit) in semi-skirted low-DNA-binding 96-well plates (Eppendorf twin.tec LoBind). Immediately after sorting, lysis and bisulfite conversion was performed by heating to 98 °C for 8 min, followed by three conversion steps (64 °C for 1 h) alternating with denaturation steps (98 °C for 2 min). One microliter of 10 M sodium hydroxide was added to facilitate desulfonation; the reaction was then incubated at 37 °C for 15 min. Ninety-six microliters of 10.4 mM Tris-HCl was then added to neutralize the reaction mixture and lower the concentration of sodium bisulfite, for a final volume of 100 µL.

Primer composition was determined by counting the proportion of 2-base kmers in the genome, then changing all Cs to Ts except in the CpG context. This yielded an estimate of 49% T, 30% A, 20% G, and 1% C (in the CpG context) in the converted genome. Thus, our first-strand random primer is composed of 50% A, 30% T, and 20% C, plus a 1% spike-in of G exclusively in CpG context; the second-strand random primer is the complement of this composition.

We performed several experiments to determine the optimal concentration of first-strand random primer and found that using more random primer resulted in excessive adapter contamination upon sequencing, while using less random primer resulted in low yields. We found the concentration reported below to be a good

balance between these challenges for euploid human and mouse cells, with minimal adapter contamination (Fig. 1c, f) and high success rates. If scDEEP-mC is applied to other inputs, such as haploid gametes or small cell pools, it may be necessary to empirically titrate the primer concentration for these samples. (Notably, we found the protocol to be less sensitive to the concentration of second-strand random primer.)

The first strand primer mix was prepared, consisting of 0.5 µM Oligo 1 and 0.625 nM each G-poor CpG spike-in primer. Next, 14 µL of first strand random priming mixture (170 µL first strand primer mix, 170 µL 10 mM dNTPs, 1166 µL 10× Blue Buffer) was added to the reaction, which was incubated at 65 °C for 3 min, then chilled to 15 °C. One microliter (50U) of high-concentration klenow 3′–5′ exo- polymerase (Qiagen) was added, and a linear amplification cycle was initiated (15 °C for 5 min, slow heating [0.1 °C/s] to 37 °C, hold at 37 °C for 5 min, chill to 15 °C for 10 s, heat to 95 °C for 1 min). This constitutes the first round of random priming. The reaction was then chilled to 15 °C, and 2.5 µL of master mix (740 µL first strand primer mix, 74 µL 10 mM dNTPs, 185 µL 10× Blue Buffer, 370 µL high-concentration klenow exo-, 481 µL water) was added before another linear amplification cycle was performed, using the same incubation steps as above. This process was repeated 5 times (random priming rounds 2–6). Finally, 5 µL of master mix was added; the reaction was chilled to 15 °C for 5 min, slowly heated (0.1 °C/s) to 37 °C, held at 37 °C for 5 min; chilled to 15 °C for 5 min, slowly heated (0.1 °C/s) to 37 °C, held at 37 °C for 90 min, and chilled to 4 °C. In this way, a total of seven rounds of random priming were completed. Prior to second-strand synthesis, 2 µL (40 U) Exo I (NEB) was added, and the reaction was incubated at 37 °C for 1 h to digest single-stranded DNA fragments. A bead cleanup (Beckman Coulter SPRIselect) was then performed using 106 µL of beads per reaction (-0.8× ratio), eluting into 40 µL of water.

To incorporate the second sequencing primer, a second strand primer mix was prepared, consisting of 10 µM Oligo 2 and 12.5 nM each C-poor CpG spike-in primer. Nine microliters of second-strand master mix (206 µL 10 mM dNTP, 206 µL s strand primer mix, 515 µL 10× Blue buffer) was then added, and the reaction was incubated at 95 °C for 1 min, then cooled to 4 °C. Next, 2 µL (100 U) of klenow exo- was added, and the reaction was held at 4 °C for 5 min before being slowly heated (0.1 °C/s) to 37 °C, then held at 37 °C for 90 min. The reaction was cooled to 4 °C and a second bead cleanup was performed, using 40 µL of beads (0.8× ratio) and eluting into 23 µL of water. 23 µL of KAPA HotStart ReadyMix master mix (Roche) and 2 µL of xGen UDI indexing primers (IDT) were added, followed by PCR amplification (95 °C for 2 min; 12–15 cycles of melting [94 °C for 80 s], annealing [62 °C for 30 s], and extension [72 °C 40 s]; 72 °C for 3 min). A final bead cleanup was performed using 35 µL of beads (0.7×) and eluting into 21 µL of storage buffer (10 mM Tris, 0.1 mM EDTA, pH 8.0).

After preparation, the concentration and fragment size distribution of each library was quantified using the Quant-iT PicoGreen dsDNA kit (Invitrogen) and High Sensitivity D1000 ScreenTape Assay (Agilent), respectively. Libraries (typically 30–35 per pool) were pooled in an equimolar fashion and the pool was purified with a 0.8× ratio bead cleanup. Libraries were sequenced targeting a depth of ~30–35 M 150 bp paired-end reads using an Illumina NovaSeq 6000, with 10% PhiX spike-in. We find limited returns sequencing scDEEP-mC libraries at depths exceeding 35 M reads per cell.

## Sequence data processing

Reference sequences (mm10 for mouse, hg38 for human) were downloaded using refgenie[32] with the default settings.

Base calls were demultiplexed using bcl2fastq, allowing for one mismatch, by the sequencing provider. FASTQ files generated for the same library on multiple lanes were merged using cat. Adapters were trimmed using cutadapt[33] with the parameters --overlap 1 -a AGATCGGAAGAGC -A AGATCGGAAGAGC --error-rate 0.1 --trim-n --minimum-length 20 --nextseq-trim 20 -n 2. Paired, trimmed reads

were aligned using Biscuit[19] in a paired directional fashion (biscuit align -b1 /path/to/ref read_2 read_1). Note that read 2 is supplied before read 1 to allow for the correct directionality in alignment. Singleton reads (in which one read of a mate pair was discarded by cutadapt) were aligned in a single-ended, directional fashion (biscuit align -b 3 /path/to/ref read_1 or biscuit align -b 1 /path/to/ref read_2). For each cell, all three alignments (paired-end, single-end read 1, single-end read 2) were merged and then duplicate marked using dupsifter[34] in single-end mode. The resulting alignments are referred to as "raw" alignments.

Raw alignments were subsequently cleaned by removing reads with more than 1 CpY retention event (using biscuit bsconv), reads with MAPQ < 40, and unmapped, secondary, duplicate, or vendor-QC-failed reads (samtools view -q 40 -F 0x704)[35]. Regions of anomalously high coverage were then identified (as below) and removed to produce "polished" alignments. These "polished" alignments were used for all downstream analyses except those shown in Fig. 1, where very strict bisulfite conversion filters were applied due to the high rate of incomplete bisulfite conversion in some datasets.

### Identification of anomalous high-coverage regions
The genome was divided into 100 bp windows, tiled every 50 bp across the genome. For each cell, the number of reads overlapping each bin by at least 50% was tallied. Regions were flagged as anomalous if the number of reads in that bin was above the 80th percentile for that cell; anomalous regions were extended until the number of reads in the bin dropped below the 67th percentile for that cell. Flagged regions ≤250 bp apart were then merged. Anomalous regions were identified as bins that were flagged in >90% of cells. Finally, anomalous regions <250 bp apart were joined together to create a list of regions to exclude.

### Methylation calling
Methylation data was extracted from "polished" alignments using biscuit pileup with the flags −5 10 −3 5 -p, then summarized using biscuit vcf2bed -t CG -k 1 and biscuit mergecg. CpN retention was calculated using biscuit qc. Subsequent analyses were performed using R (version 4) as detailed below.

### Quality control
Quality evaluation was performed using FastQC[36] (on raw and trimmed FASTQ files) and biscuit qc (on raw and polished alignments). MultiQC[37] was used to summarize QC results. Low-efficiency cells were excluded if they covered fewer than 20% of CpGs or were sequenced to >55M reads (human) or >80M reads (mouse).

### Comparative analysis
Publicly available data were downloaded from the SRA and NGDC. Only cells from normal tissue from Bian et al.[7] were considered, due to possible hyperdiploidy or whole-genome doubling in tumor cells. A random sample of 10% of the Cao et al. dataset[18] was considered due to the very large size of the dataset. fastqc and multiqc were used to assess adapter content. Adapters were trimmed using cutadapt with the parameters --error-rate 0.1 --trim-n --minimum-length 20 -n 2 --overlap 1, either --quality-cutoff 20 or --nextseq-trim 20 (if the data was generated on a sequencer using 2-color chemistry), and additional parameters as described in Table 1.

Trimmed fastqs were aligned and duplicate-marked as described above, but non-directional libraries (scBS-seq, scM&T-seq, scTrio-seq, PBAL, and Cabernet) were aligned without use of the -b flag, as shown in Table 2.

Per-protocol identification of anomalous high-coverage regions was performed as described above. Subsequently, aligned reads were filtered as described above, with no protocol-specific changes.

Methylation calling was performed on "polished" alignments using biscuit pileup with the -p flag set. Parameters were set on a per-

protocol basis to mitigate the effects of random-priming-induced bias in methylation calls, as shown in Table 3.

### Data sparsity
Single-cell sequencing (and DNA methylation) data is often incomplete or sparse. Although scDEEP-mC facilitates high coverage compared to many existing methods, the resulting data is still sparse in nature, especially the ARM data (since allele assignment requires a heterozygous SNP near the CpG). The methods described below detail how this sparsity is handled in each analysis, but it may be helpful to describe some overarching principles which guided our analysis.

We do not make use of imputation, since this does not add any truly new information and may introduce noise (e.g., training data gathered from datasets with limited relevance). We also strive to avoid summarizing methylation values over large genomic bins that are defined without reference to underlying biology (e.g., mean beta over 100 kb bins). Rather, we reason that a few high-quality, informative sites with high biological significance are preferable to a large number of noisy or uninformative sites. Thus, we strive to restrict our analyses to exactly the CpGs of interest. These methylation calls are then often summarized per-region or per-cell, discarding regions or cells with insufficient data. In this way, we allow for missing data and non-overlapping coverage between cells without obscuring underlying biological differences. For dimension

**Table 1 | Trimming parameters used for the protocols compared in this study**

| Protocol | Trimming parameters |
| --- | --- |
| Cabernet (Xie) | -a CTGTCTCTTATA<br>-A CTGTCTCTTATA |
| Laird PBAL (Hirst) | -a AGATCGGAAGAGC<br>-A AGATCGGAAGAGC |
| snmC-seq2 (Ecker) | -U 14<br>-a AGATCGGAAGAGC<br>-A AGATCGGAAGAGC |
| scBS-seq (Kelsey)<br>scM&T-seq (Reik)<br>scTrio-seq (Fu) | -a AGATCGGAAGAGC<br>-A AGATCGGAAGAGC<br>-a CTACACGACGCTCTTCCGATCT<br>-A CTACACGACGCTCTTCCGATCT |

**Table 2 | Alignment parameters used for the protocols compared in this study**

| Protocol | Directional? | Read 1 mapped to: |
| --- | --- | --- |
| Laird<br>snmC-seq2 (Ecker) | Yes | Daughter strand |
| scBS-seq (Kelsey)<br>scM&T-seq (Reik)<br>scTrio-seq (Fu)<br>PBAL (Hirst)<br>Cabernet (Xie) | No | N/A |

**Table 3 | Methylation calling parameters used for the protocols compared in this study**

| Protocol | Min. dist. to 5′ end (−5 flag) | Min. dist. to 3′ end (−3 flag) |
| --- | --- | --- |
| Laird | 10 | 5 |
| snmC-seq2 (mouse) | 10 | 7 |
| snmC-seq2 (human) | 10 | 3 |
| scM&T-seq | 7 | 5 |
| scTrio-seq | 10 | 5 |
| scBS-seq | 10 | 3 |
| PBAL | 7 | 2 |
| Cabernet | 10 | 2 |

reduction, we prefer NMF, as implemented by the RcppML package[38], since it natively accommodates missing values through sparse matrices.

## Methylation discordance

Methylation discordance was calculated as follows. For each cell, CpG sites covered more than once were classified as concordant ($\beta$ exactly 0 or 1) or discordant ($0 < \beta < 1$). Then, the "discordance metric" was calculated as

$$D = \frac{\sum_{i=2}^{N} \frac{d_i 2^i}{2^i - 2}}{\sum_{i=2}^{N} n_i} \qquad (1)$$

where $D$ is discordance, $d_i$ is the number of discordant sites at coverage depth $i$, and $n_i$ is the total number of sites covered $i$ times. Intuitively, this can be conceptualized as calculating the proportion of discordant sites at a given coverage depth (normalized by the probability of observing discordance given random assortment of methylation calls), then taking the mean across all observed depths (weighting by the number of observations at that depth). $D$ may be calculated for either allele-resolved or summarized methylation data. In the former case, $D$ represents the accumulation of polymerase, sequencing, and allele assignment errors, while in the latter case, $D$ also incorporates hemi-methylation and allele-specific methylation.

## Doublet identification

Libraries generated from K562 cells and tumor samples were excluded, as we found that their hypotriploid karyotype or copy number alterations, respectively, produced artificially inflated coverage due to an increased amount of input DNA. scDEEP-mC libraries were excluded if their allele-resolved CpG discordance rate was >2%. Libraries from other protocols were excluded if they were not derived from embryonic stem cells and had a CpG discordance rate >15%.

## Bisulfite conversion

Base-resolution CpA, CpC, and CpT retention rates were calculated using biscuit vcf2bed. CpY retention was calculated as the mean of CpC and CpT retention rates.

## Sequencing efficiency

Adapter trimming yield was computed by calculating the total number of bases in the trimmed fastq files, divided by the expected number of sequenced bases (reported sequenced reads * read length [* 2 if paired]). For subsequent steps, 2 million (paired-end, if applicable) reads were randomly sampled from each fastq file using seqtk[39]. These reads were extracted from the aligned .bam file, and duplicate flags were unset. The number of aligned bases in the subset reads was tallied to calculate the alignment yield. Duplicates were then re-marked (using dupsifter, as above) in each subset of reads. Reads with 1 or more CpY retention events (strict filter) were then discarded (using biscuit bsconv, as above). Next, unmapped, vendor-QC-fail, secondary, and low-mapping-quality alignments were discarded (samtools view -F 0×304 -q 40). Next, duplicate reads were discarded. After each filtering step, the number of bases in the remaining filtered alignments was tallied using samtools depth. Finally, the number of bases covered by the filtered reads was tallied using samtools coverage.

## Coverage comparisons

For Fig. 1d, g, 20M reads were randomly sampled from each cell. If a cell had less than 20M total reads, all available reads were considered. Selected reads were extracted from the aligned .bam file and duplicates were re-marked as above. Reads with 1 or more CpY retention events (strict filter) were then discarded, as were unmapped, vendor-QC-fail, secondary, and low-mapping-quality alignments (as above). Finally, reads mapping to regions of anomalously high coverage (as

defined above) were discarded, and coverage was measured using samtools coverage.

## Cell-type-specific hypomethylated regions

A table of 50286 cell type-specific unmethylated markers (top 1000 for each cell type) was downloaded from ref. 20. These regions were lifted over from the hg19 source genome to the mm10 genome using the liftover tool and hg19ToMm10.over.chain.gz file provided by UCSC. For each cell, the mean $\beta$ value of all covered CpGs inside all regions specific to a particular cell type (or group of cell types) was calculated. The mean $\beta$ values for each cell for the "Blood-T", "Blood-B", "Colon-Ep:Gastric-Ep:Small-Int-Ep", "Skeletal-Musc:Smooth-Musc", "Blood-Mono+Macro", and "Colon-Fibro" regions were extracted, and cells and regions were hierarchically clustered based on mean $\beta$ values using Ward's algorithm (method = "ward.D2"). Data was visualized using the ComplexHeatmap package[40]. Cells were classified based on their mean $\beta$ values for the "Blood-T" and "Colon-Ep:Gastric-Ep:Small-Int-Ep" cell types, based on the following criteria:

| | Gastrointestinal epithelial mean $\beta > 0.65$ | Gastrointestinal epithelial mean $\beta < 0.65$ |
|---|---|---|
| "Blood-T" mean $\beta > 0.65$ | Other | GI epithelial |
| "Blood-T" mean $\beta < 0.65$ | Lymphocyte | Doublet of two different cell types |

## Solo-WCGW methylation

Solo-WCGW CpGs in common PMDs in the mm10 and hg38 genome were downloaded from ref. 9. The mean $\beta$ value over all covered CpGs in these regions was calculated for each cell.

## Promoter methylation

For each gene in the GENCODE VM23 (UCSC knownGene) annotation, the region 1.5 kb upstream and 200 bp downstream of the start site was taken as the promoter. The mean $\beta$ value of all covered CpGs in each promoter was calculated for each cell. For visualization, promoters with >25% of CpGs covered in ≥60% of cells, and having a standard deviation of >0.2 across all cells, were selected. Cells and promoters were hierarchically clustered based on mean $\beta$ values using Ward's algorithm. For differential methylation analysis, the cells were classified as "immune" or "epithelial" as described under "Cell-type-specific hypomethylated regions". An unpaired two-tailed $t$-test was used to compare mean $\beta$ values of each promoter in all cells in each group; $p$ values were corrected using the Benjamini-Hochberg algorithm. Promoters overlapping CpG islands (as defined by ref. 41) by at least 200 bp were annotated appropriately.

## Non-negative matrix factorization

For each CpG, the number of cells covering that CpG and the standard deviation (SD) of $\beta$ value across all cells were calculated. CpGs above the 25th percentile for coverage and 67th percentile for SD were selected for NMF analysis. $\beta$ values for each cell were accumulated into a sparse vector; a complementary "unmethylation" vector (1-$\beta$) was appended to this vector to prevent subsequent normalization from being affected by methylation state. Since the R package "Matrix" represents missing values as 0 in a sparse vector, $\beta$ values of 0 were replaced with $1 \times 10^{-12}$ and missing values were replaced with 0. The vectors were concatenated as the columns of a sparse matrix, which was column-normalized and log-transformed. The resulting matrix was subjected to rank−2 NMF, masking zero (missing) values, as implemented by the RcppML package[38].

## Variant discovery from scWGBS data

Due to the nature of scDEEP-mC (and other methods involving multiple rounds of random priming), it is possible for multiple library molecules to be generated from the same fragment. This results in a situation where both first-in-pair reads have the same start position, but their mates have differing start points. Generally, duplicate marking tools would not flag these pairs as duplicates. However, we found that including such read pairs led to a higher rate of false-positive variant calls. Thus, for the purposes of variant calling, we adopted a stricter duplicate marking approach by first "unpairing" all reads (unsetting all mate-related flags and tags, while tagging the read name with the read number), then re-marking duplicates in single-end mode. These strictly-deduplicated reads were merged for pseudo-bulk variant calling or taken separately for single-cell variant calling. All variant calling was performed using biscuit pileup with the flags -5 10 -3 5 -p -n 7. Regions that harbored an unusual amount of low-mapping-quality reads also yielded many false-positive variant calls. To identify these regions, the mean mapping quality of all strictly-deduplicated reads was calculated for non-overlapping 500 bp bins, and bins with a mean MAPQ < 45 were removed from further consideration. Variants were selected if they had an allele frequency between 0.25 and 0.75 in pseudo-bulk analysis, as well as a quorum of cells (≥5 cells or ≥5% of cells, whichever was greater) directly supporting each variant call. As expected, a substantial number of SNPs were at CpG sites. SNPs that create or destroy CpGs could lead to a methylation bias in the results. To mitigate this, we identified SNPs where the alternate allele created a CpG and used bcftools consensus to "apply" these alternate alleles to the references, thus creating a new reference genome with all possible CpGs to use for methylation calling.

## Curation of mouse SNPs

We collected tail clippings for 4 siblings (2 male, 2 female) of the CB6F1/J mouse from which we generated scDEEP-mC libraries (Jackson Labs Strain # 100007). Genomic DNA was extracted from this tissue using the Qiagen DNeasy Blood & Tissue Kit. Libraries for WGS were prepared by the Van Andel Genomics Core from 500 ng of high molecular weight genomic DNA using the KAPA Hyper Prep Kit (v5.16) (Kapa Biosystems, Wilmington, MA USA). In brief, DNA was sheared to an average size of 450 bp. Then, end-repaired and A-tailed DNA fragments were ligated to uniquely barcoded dual indexes (IDT, Coralville, IA USA), after which minimal PCR (6 cycles) was performed. Whole-genome libraries were sequenced to a depth of 30× on Illumina NovaSeq X.

Alignment (to GRCm38) and variant calling was performed using the nf-core/sarek pipeline (version 3.4.2), using haplotypecaller, bcftools, freebayes, and strelka to call variants. Variants were filtered using appropriate cutoffs for each tool (haplotypecaller: quality > 100, bcftools: qual > 221, freebayes: qual > 25, strelka: qual > 25, and passing filters). Variants called by at least 3 callers were selected; selected variants found in at least 3 out of 4 mice (autosomes) or 2 out of 2 female mice (chromosome X) were selected. Finally, selected variants were filtered to only include those found in the Mouse Genome Project variant catalog for BALB/c.

## Local variant phasing

It is possible for reads to contain more than one high-quality heterozygous SNP. Thus, it is important to ensure that SNPs are not randomly assigned to alleles, which could introduce unnecessary conflicts when trying to assign reads to alleles. To address this problem, we used whatshap, a read-backed phasing algorithm[15], to ensure that variants were assigned to alleles in a locally consistent fashion. However, whatshap is not designed with bisulfite sequence data (and its attendant ambiguities) in mind. To overcome this issue, we "deconverted" our reads using a custom Julia script. This script performs the following actions:

- For bases that are marked as insertions relative to the reference, ambiguous bases (T on the C- > T converted strand; A on the G- > A converted strand) are replaced with N
- For bases that are at a known, high-quality heterozygous SNP, ambiguous bases are replaced with N and informative bases are replaced with the supported base. For example, a C or T is ambiguous if it is on the C- > T converted strand at a C/T SNP. However, a T base at an A/C SNP supports the C allele, and is replaced with a C.
- All other bases are replaced with the reference sequence

Unpaired, strictly-deduplicated, "deconverted" reads were then used to conduct local SNP phasing using whatshap phase with the parameters --only-snvs --ignore-read-groups --merge-reads with the appropriate genome as a reference. The output of whatshap phase was further edited such that singleton SNPs (which could not be phased into larger blocks) were transformed into singleton phase groups containing that single SNP.

## Assignment of reads to local haplotypes

The 5′ and 3′ ends of reads may have substantial methylation bias due to imperfect priming. Thus, we manually trimmed reads using trimBam (parameters -L 10 -R 5)[42] and used samtools view to exclude reads with 7 or more mismatches. Filtered, "deconverted" reads were then assigned to alleles using whatshap haplotag and the locally phased, high-quality heterozygous SNPs previously generated. The allele assignments generated by this tool were transferred to the original reads using a custom Julia script.

## Allele-resolved methylation calls

Reads assigned to each allele were extracted using samtools view. Methylation was called against an updated reference containing all possible CpGs (including those created by SNPs), as described above. Methylation calling was performed as above, using biscuit pileup and biscuit vcf2bed. Finally, a custom Python script was used to annotate the output file with the phase sets that contributed to each allele-specific methylation call. Methylation calls were discarded in the following situations: 1) Methylation call at a CpG-creating SNP on the allele where there is no CpG; 2) reads from multiple phase sets contributed to the methylation call; or 3) methylation calls at CpGs where both bases were SNPs. Per-read SNP and methylation call data were visualized using the R packages Rsamtools[43], Biscuiteer[44], and Bisplotti[45].

## Transcription factor binding site hemi-methylation

The TFBS cataloged in the LOLA[46] Core "codex" and "encodeTFBS" databases for the mm10 genome were used for this analysis. Two-strand sites (CpGs where both the top and bottom strand were covered, and both strands were confidently assigned to the same microhaplotype) were selected and grouped by their methylation state (symmetrically unmethylated, hemi-methylated, or symmetrically unmethylated). Finally, the fraction of sites in each state overlapping each TFBS region set was calculated; TFBS region sets with ≤250 methylation or hemi-methylation events were discarded.

## Read count analyses

The number of "polished" alignments falling within 50 kb non-overlapping bins was tallied using bedtools[47], then normalized by the total number of polished, aligned reads per cell to yield reads per million mapped (RPMM). Per-bin RPMM biases were normalized as follows: Bins with RPMM <5th or > 95th percentile for their cell, in >75% of cells, were excluded, as were bins with >10% missing data. "Ideal" cells with a uniform read distribution were selected by computing the standard deviation (SD) of RPMM in 5 Mb non-overlapping

bins. Bins with an SD < 2.5 were selected, and the standard deviation of the mean read count over all bins was calculated for each cell. Cells with a whole-genome SD between 1 and 1.75 were flagged as "ideal" cells. A per-bin correction factor was obtained by calculating the median RPMM for each 50 kb bin across all ideal cells; RPMM counts across all cells were normalized by these bin-specific corrections.

## Replication timing

High-resolution repli-seq data was downloaded from ref. 26. In this dataset, 50 kb non-overlapping bins tiling the genome are assigned 16 scores, corresponding to the normalized read depth in cells collected in 16 fractions through S-phase. We assigned each 50 kb bin a replication timing score (from 1 to 16) by identifying the three highest-scoring fractions for each bin. Then, we took the mean of these fraction indices, weighted by their scores, and rounded to the nearest integer. Replication bins 1, 15, and 16 were excluded from further analysis, as they encompassed disproportionately large or small regions of the genome and may have introduced bias. For each replication timing bin, we tallied two-strand methylation states (symmetric methylation, hemi-methylation, or symmetric unmethylation) from scDEEP-mC data across all G1-phase cells in each passage (Fig. 4a).

Replication timing bins were further grouped as follows: "very early" included bins 2–4; "early" included bins 5–8; "late" included bins 9–12; and "very late" included bins 13–14. Cells with > 33.5% of normalized reads in the "very early" bin and <3% allele-resolved CpG discordance were labeled as "S-phase" cells (Fig. 4d).

For Fig. 4e, the normalized read count was converted to estimated ploidy by calculating the mode of the normalized read count density. Normalized read counts were divided by this mode, then multiplied by 2 (if the mode was ≤1.1) or 4 (if the mode was >1.1, or the cell was flagged as G2 phase). The genome was partitioned into 50 kb bins, and the $\beta$ values for all CpGs covered ≥2× were discretized into three groups: unmethylated ($\beta = 0$), methylated ($\beta = 1$), and partially methylated ($0 < \beta < 1$). Bins with <15 methylation measurements were discarded. The remaining bins were grouped by their estimated ploidy and the total proportion of CpGs in each state was calculated. The opacity of each bin was scaled by its weight, which was calculated as $\frac{sum\ of\ RPMM\ in\ bin}{sum\ of\ RPMM\ for\ cell}$.

## X-inactivation analysis

Per-allele mean β values of CpGs within CpG islands (as defined by ref. 41) were calculated per cell (Fig. 5a); cells with <50 methylation calls for each allele were removed. For Fig. 5b, three region sets were considered. *CpG island promoters* were defined as CpG islands within 1.5 kb upstream−1 kb downstream of a transcription start site, with the remaining CpG islands being described as *non-promoter CpG islands*. *Non-CpG island promoters* were defined as regions 1.5 kb upstream−200 bp downstream of genes, which did not have a CpG island 1.5 kb upstream−1 kb downstream of the TSS. All CpGs within a TSS, or within a CpG island/shores (±500 bp), were averaged across all cells (considering each cell's X-inactivation state) to produce a mean β for each feature on both the active and inactive X chromosomes. Features with <100 observations (methylation calls) were discarded.

For Fig. 5c, gene promoters (both CGI and non-CGI) with ≥ 100 methylation calls, a mean $\beta < 0.25$ on the active allele, and a mean $\beta > 0.25$ on the inactive allele were considered. The positions of CpGs within these genes (and the regions 5 kb upstream and downstream) were normalized to align to each other, then binned. Mean β values were calculated for the active and inactive allele within each bin; bins within genes were weighted by the length of the gene (since shorter genes have fewer measurements, and thus more uncertainty). Mean β values were smoothed using a loess algorithm with span = 0.2 for final visualization.

## NMF of chromosome X methylation data

All ARM calls on chromosome X were considered for NMF. $\beta$ value complementation, zero-masking, log-normalization, and factorization were performed as described above. After factorization, features >500 bp away from a CpG island were excluded. NMF features having either factor value >95th percentile for that factor were excluded. Subsequently, factor loadings were scaled to a maximum of 1, and features with a scaled loading <0.05 or >0.85 (0.75 for human fibroblasts) in both factors were excluded. Features were assigned to whichever factor had a greater scaled loading. Since each locus is represented by four features (one for each allele, one for methylation and one for its complement), CpGs were assigned to NMF-derived alleles by majority vote, discarding CpGs with ties.

## Human fibroblast chromosome X analysis (Fig. 6)

Methylation calls on chromosome X from early-passage cells were subjected to NMF as described above. For each cell, the number of methylation calls assigned to each NMF factor (methylation-derived allele) was tallied to obtain the allele copy number. Likewise, the real $\beta$ values were summarized to compute allele methylation. Feature assignments derived from NMF analysis of early-passage cells were used to "phase" methylation calls from late- and very late-passage cells.

Normalized read counts were obtained as described above. Cells with <24% of reads in the "very early" replication timing bin, or <8 M CpGs covered, were discarded (due to high variability in read distribution). Normalized read counts were binned into 0.5 Mb non-overlapping bins and averaged for visualization. Cells were grouped using hierarchical clustering with Ward's algorithm. The three major clusters in very-late-passage cells were identified and $\beta$ values in solo-WCGW CpGs in common PMDs[9] were summarized (as above).

A mean β value was calculated for each CpG island in each cell, discarding CpG islands with ≤30% of CpGs covered. CpG islands with a standard deviation >0.2 across all cells, and <20% missing data, were selected as "variable". Next, tow-tailed *t*-tests were computed for each CpG island and major late-passage cluster, comparing its methylation in the cluster to the other two clusters. CpG islands with a Benjamini-Hochberg-adjusted *P*-value < 0.01 and a mean delta $\beta > 0.15$ were selected as "differentially methylated". Cells were hierarchically clustered based on their mean $\beta$ values at selected CpG islands, and CpG islands were arranged in order of increasing mean $\beta$ across all very-late-passage cells.

## Statistics and reproducibility

This study was designed to illustrate the strengths of scDEEP-mC and related analyses in a variety of contexts. As such, the experiments were not randomized, and the Investigators were not blinded to allocation during experiments and outcome assessment. No statistical method was used to predetermine sample size, since we were not searching for a rare cell population. Cell numbers were chosen to capture major populations in the sample of interest and facilitate rapid experimental iteration. Doublet libraries were identified based on methylation discordance and removed as described above. Statistical analyses were conducted using R (version 4.4). Bioinformatics workflows were implemented using a containerized Nextflow pipeline (version 24.04) to allow complete reproducibility.

## Primer sequences

All primers were ordered from IDT; the sequences given below are given in IDT ordering format, with the random primer sequence underlined.

**First-strand primers.** Oligo 1:
ACACTCTTTCCCTACACGACGCTCTTCCGATCT(N1:50200030)(N1)(N1)(N1)(N1)(N1)(N1)(N1)

G-poor CpG spike-in 1:

ACACTCTTTCCCTACACGACGCTCTTCCGATCT<u>CGHHHHHHH</u>
G-poor CpG spike-in 2:
ACACTCTTTCCCTACACGACGCTCTTCCGATCT<u>HCGHHHHHH</u>
G-poor CpG spike-in 3:
ACACTCTTTCCCTACACGACGCTCTTCCGATCT<u>HHCGHHHHH</u>
G-poor CpG spike-in 4:
ACACTCTTTCCCTACACGACGCTCTTCCGATCT<u>HHHCGHHHH</u>
G-poor CpG spike-in 5:
ACACTCTTTCCCTACACGACGCTCTTCCGATCT<u>HHHHCGHHH</u>
G-poor CpG spike-in 6:
ACACTCTTTCCCTACACGACGCTCTTCCGATCT<u>HHHHHCGHH</u>
G-poor CpG spike-in 7:
ACACTCTTTCCCTACACGACGCTCTTCCGATCT<u>HHHHHHCGH</u>
G-poor CpG spike-in 8:
ACACTCTTTCCCTACACGACGCTCTTCCGATCT<u>HHHHHHHCG</u>

**Second-strand primers.** Oligo 2:
GTGACTGGAGTTCA-
GACGTGTGCTCTTCCGATCT<u>(N2:30002050)(N2)(N2)(N2)(N2)(N2)(N2)(N2)</u>

C-poor CpG spike-in 1:
GTGACTGGAGTTCAGACGTGTGCTCTTCCGATCT<u>CGDDDDDDD</u>
C-poor CpG spike-in 2:
GTGACTGGAGTTCAGACGTGTGCTCTTCCGATCT<u>DCGDDDDDD</u>
C-poor CpG spike-in 3:
GTGACTGGAGTTCAGACGTGTGCTCTTCCGATCT<u>DDCGDDDDD</u>
C-poor CpG spike-in 4:
GTGACTGGAGTTCAGACGTGTGCTCTTCCGATCT<u>DDDCGDDDD</u>
C-poor CpG spike-in 5:
GTGACTGGAGTTCAGACGTGTGCTCTTCCGATCT<u>DDDDCGDDD</u>
C-poor CpG spike-in 6:
GTGACTGGAGTTCAGACGTGTGCTCTTCCGATCT<u>DDDDDCGDD</u>
C-poor CpG spike-in 7:
GTGACTGGAGTTCAGACGTGTGCTCTTCCGATCT<u>DDDDDDCGD</u>
C-poor CpG spike-in 8:
GTGACTGGAGTTCAGACGTGTGCTCTTCCGATCT<u>DDDDDDDCG</u>

**Reporting summary**

Further information on research design is available in the Nature Portfolio Reporting Summary linked to this article.

## Data availability

Raw scDEEP-mC sequencing data and processed methylation call data has been deposited in the Gene Expression Omnibus (GSE280161). Figure source data is provided with this paper. The following public datasets used in this study are also available from the Gene Expression Omnibus: scBS-seq[3] data (GSE56879), single-cell PBAL[4] data (GSE89545), scM&T[5] data (GSE68642), snmC-seq2[6] data (GSE112471), and scTrio-seq[7] data (GSE97693). Cabernet[18] data is available from the Genome Sequence Archive (CRA005812). Source data are provided with this paper.

## Code availability

Analysis code used in this study is deposited in Zenodo, and can be accessed using the https://doi.org/10.5281/zenodo.15596168.

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

## Acknowledgements

This work was supported by NIH grants R01CA234125 and R01AG084743 awarded to P.W.L., and R37CA230748 awarded to H.S. We thank Zach DeBruine for insightful discussions regarding NMF. We are grateful to Liang Kang and Emily Jung for assistance with animal husbandry and cell culture experiments. We thank the Vivarium (RRID:SCR_023211), Flow Cytometry (RRID:SCR_022685) and Genomics Core (RRID:SCR_022913) resources at Van Andel Institute for their contributions of technical expertise and insights. Computation for the work described in this paper was supported by the High Performance Cluster and Cloud Computing (HPC3) Resource directed by Zach Ramjan at the Van Andel Institute.

## Author contributions

N.J.S. contributed to conceptualization, methodology, software, validation, formal analysis, investigation, data curation, writing—original draft, writing—review and editing, visualization. W.A.H. and Z.Z. contributed to conceptualization, methodology, validation, investigation. E.E., H.Y.M., D.S., and K.H.L. contributed to methodology, investigation. P.A.N. and J.L.E. contributed to investigation, resources. K.F.K. contributed to resources, project administration. T.H., J.M., B.K.J., and W.Z. contributed to resources, software. H.S. contributed to conceptualization, methodology, supervision, and funding acquisition. P.W.L. contributed to conceptualization, methodology, writing—review and editing, supervision, visualization, project administration, funding acquisition.

## Competing interests

P.W.L. serves on the scientific advisory board of Tagomics. P.W.L. and H.S. both serve on the scientific advisory board of FOXO Technologies. The remaining authors declare no competing interests.

## Additional information

**Nathan J. Spix** ®[1,6], **Walid Abi Habib** ®[1,2,6], **Zhouwei Zhang**[1,3], **Emily Eugster**[1], **Hsiao-yun Milliron**[1], **David Sokol**[1], **KwangHo Lee**[1], **Paula A. Nolte**[1], **Jamie L. Endicott**[1,4], **Kelly F. Krzyzanowski**[1], **Toshinori Hinoue**[1], **Jacob Morrison**[1], **Benjamin K. Johnson**[1], **Wanding Zhou** ®[1,5], **Hui Shen** ®[1] ✉ **& Peter W. Laird** ®[1] ✉

[1]Department of Epigenetics, Van Andel Institute, Grand Rapids, MI, USA. [2]Present address: Takara Bio Europe, Saint-Germain-en-Laye, France. [3]Present address: McKinsey & Company, Boston, MA, USA. [4]Present address: Altos Labs, San Diego, CA, USA. [5]Present address: University of Pennsylvania, Philadelphia, PA, USA. [6]These authors contributed equally: Nathan J. Spix, Walid Abi Habib. ✉e-mail: hui.shen@vai.org; peter.laird@vai.org

