## [Peer Review file · Nature Communications]

High-coverage allele-resolved single-cell DNA methylation profiling reveals cell lineage, X-inactivation state, and replication dynamics

Corresponding Author: Dr Peter W. Laird

Version 0:

Reviewer comments:

Reviewer #1

(Remarks to the Author)

The authors present scDEEP-mC, an improved method for single-cell whole-genome methylation sequencing. They applied scDEEP-mC to primary mouse intestinal cells and human fibroblasts, benchmarking it against existing scWGBS methods. Their results demonstrate superior coverage of scDEEP-mC. Using the mouse intestinal scDEEP-mC data, they performed cell-type annotation and identified differentially methylated genes, including known marker genes. Furthermore, they developed an allele-resolved methylation analysis pipeline that, leveraging the higher coverage of scDEEP-mC, detects both allelic methylated and hemi-methylated regions. They observed a higher frequency of hemi-methylation at certain TFBSs, including those of the pioneer transcription factor Sox2. They also investigated the relationship between DNA replication timing and demethylation, and, without relying on phased SNPs, performed X-inactivation analysis using the scDEEP-mC data, examining changes in X-inactivation during long-term culture. Given the performance of scDEEP-mC and the diverse biological applications demonstrated, this study is significant and will likely be of broad interest. However, I have a few minor comments:

1. It might be a problem of PDF conversion; there is a space in 'contrast' in line 128.
2. While the benchmarking of scDEEP-mC against other methods focuses on coverage, sequencing yield, and conversion rate, the reproducibility of CpG methylation status measurement requires further evaluation. Assessing the correlation of CpG methylation rates among cells, and comparing these correlations with those obtained using other methods, would provide a more comprehensive assessment of reproducibility. Additionally, an evaluation of GC bias in the sequenced regions would be beneficial to assess the uniformity of read distribution.
2. Showing known maternal and paternal methylated sites in Figure 2d would enhance its interpretability.
3. Providing supplementary tables listing the identified AMRs and hemi-methylated regions would greatly benefit the community and enhance the value of the manuscript.
4. Visualization of the methylation status of the hemi-methylated region around the TFBS using a genome browser as an example would better illustrate the finding.
5. Many readers may be unfamiliar with solo-WCGWs. Therefore, a clear explanation of solo-WCGWs and the rationale for focusing on them is necessary.

(Remarks on code availability)

Reviewer #2

(Remarks to the Author)

Comments for the manuscript entitled “high-coverage allele-resolved single-cell DNA methylation profiling reveals cell lineage, X-inactivation state, and replication dynamics” by Spix et al.

The manuscript presents a significant technical advance in single-cell whole-genome bisulfite sequencing through the introduction of scDEEP-mC. The authors convincingly demonstrate that their method achieves high library complexity and CpG coverage, overcoming key limitations of previous approaches. Notably, the technique enables allele-resolved methylation analysis, detailed interrogation of hemi-methylation, and simultaneous measurement of replication dynamics. These capabilities open new avenues for investigating cell lineage, X-inactivation, and methylation maintenance. Overall, the manuscript offers a significant methodological advance with clear implications for single-cell epigenomics. I believe this work is suitable for publication in Nature Communications following clarification and streamlining of some methodological details, as outlined below:

1. Including additional quantitative comparisons—such as reproducibility metrics across independent replicates and performance across diverse cell types—would further strengthen the claims regarding the method’s scalability and general applicability.
2. While the manuscript outlines the computational pipeline, additional details on how data sparsity is managed would benefit readers who are less familiar with single-cell bioinformatics.
3. Figures 2a and 2g are challenging to interpret. Enhancing the figure legends or adding more detailed descriptions in the main text would improve reader understanding of these key data presentations.
4. I did not find an explanation for Figure 4b in the main text. Please ensure that all figures are adequately described and integrated into the manuscript narrative.
5. A discussion of the limitations of scDEEP-mC compared with other published methods would be valuable. Specifically, addressing any potential drawbacks or trade-offs inherent to the approach will help readers contextualize the advance.

(Remarks on code availability)

Reviewer #3

(Remarks to the Author)

This manuscript presents scDEEP-mC, a promising technology for generating high-coverage single-cell DNA methylation sequencing data. The authors demonstrate its utility by conducting allele-resolved methylation analysis and validating key epigenetic phenomena. However, several concerns require further clarification:

- (1). In Figure 1, The comparison of scDEEP-mC to other methods appears to be confounded by significant differences in total number of reads per cell. Please provide a summary table or figure showing the median number of reads per cell for each dataset before downsampling. This will allow for a more accurate assessment of method performance and avoid potential biases introduced by sequence depth.
For example, If scDEEP-mC has a significantly higher total number of reads (> 20M per cell), while other method has <2M reads/cell, then, the improved performance (genome coverage) could be attributed to the increased sequencing depth rather than inherent methodological advantages.
- (2). What's the minimal coverage required for accurate allele-resolved methylation analyses and X-inactivation studies in single cells. This information would provide valuable guidance for future studies and experimental design.
- (3). mCH methylation levels vary across tissues (e.g., high (~ 6%) in neurons, low in other cell types). However, the impact of this variation on scDEEP-mC performance is not adequately addressed. Given that the initial primer design assumed negligible mCH methylation (only 1% C in CpG context during the design of random primer), its applicability to tissues with high mCH levels, such as brain, may be limited. The authors should discuss the potential implications of this assumption on the accuracy and robustness of scDEEP-mC in different tissues, especially the mCH in brain.
- (4). The manuscript lacks a comprehensive assessment of the performance of the DNA methylation phasing pipeline. Metrics such as accuracy, precision, and recall should be evaluated and presented to demonstrate the reliability of this pipeline.
- (5). The description of the methods used for Figure 4 is unclear. It is stated that data was downloaded from reference 27 (line 343) in the method part, but the analysis seems to involve data generated using scDEEP-mC (line 243) in the result part. Please clarify which analyses were performed on the downloaded dataset and which utilized data generated by scDEEP-mC.

Minor:

(1). line 247, "higher rates of hemi-methylation in early passage cells", should be "higher rates of hemi-methylation than early passage cells"?

(Remarks on code availability)

The link to the code is not accessible.

Version 1:

Reviewer comments:

Reviewer #1

(Remarks to the Author)

The authors have addressed all of my comments.

(Remarks on code availability)

Reviewer #2

(Remarks to the Author)

The authors have thoroughly addressed all of my concerns, and I have no further comments or suggestions.

(Remarks on code availability)

Reviewer #3

(Remarks to the Author)

The Authors have addressed all of my concerns.

(Remarks on code availability)

Response in Blue Italic Font

Reviewer's comments:

Reviewer #1 (Remarks to the Author):

The authors present scDEEP-mC, an improved method for single-cell whole-genome methylation sequencing. They applied scDEEP-mC to primary mouse intestinal cells and human fibroblasts, benchmarking it against existing scWGBS methods. Their results demonstrate superior coverage of scDEEP-mC. Using the mouse intestinal scDEEP-mC data, they performed cell-type annotation and identified differentially methylated genes, including known marker genes. Furthermore, they developed an allele-resolved methylation analysis pipeline that, leveraging the higher coverage of scDEEP-mC, detects both allelic methylated and hemi-methylated regions. They observed a higher frequency of hemi-methylation at certain TFBSs, including those of the pioneer transcription factor Sox2. They also investigated the relationship between DNA replication timing and demethylation, and, without relying on phased SNPs, performed X-inactivation analysis using the scDEEP-mC data, examining changes in X-inactivation during long-term culture. Given the performance of scDEEP-mC and the diverse biological applications demonstrated, this study is significant and will likely be of broad interest. However, I have a few minor comments:

1. It might be a problem of PDF conversion; there is a space in 'contrast' in line 128.

Response: Corrected – thank you for your attention to detail!

2. While the benchmarking of scDEEP-mC against other methods focuses on coverage, sequencing yield, and conversion rate, the reproducibility of CpG methylation status measurement requires further evaluation. Assessing the correlation of CpG methylation rates among cells, and comparing these correlations with those obtained using other methods, would provide a more comprehensive assessment of reproducibility. Additionally, an evaluation of GC bias in the sequenced regions would be beneficial to assess the uniformity of read distribution.

Response:

Assessment of the reproducibility of CpG methylation status measurement would ideally employ the measurement of independent replicates. However, this is not feasible for single-cell studies in which each cell is destroyed during analysis, and can therefore only be measured once. Replicate studies would by necessity involve a comparison between different cells, which introduces biological variation in addition to technical noise. An alternative approach suggested here by this reviewer is to assess the correlation of CpG methylation rates among cells. We anticipate that the degree of biological variation among primary cells obtained directly from a mouse would be higher than among fibroblasts in culture. We would further expect that correlation among fibroblasts would be highest within the same passage, and somewhat lower between passages. We have added a Supplementary Figure (Fig. S3) showing pairwise concordance between all human cells (Fig. S3a), and all mouse cells (Fig. S3b). We chose to measure concordance (fraction of sites with identical values) rather than correlation, as correlation can be destabilized by (mostly) binary data.

We considered measuring pairwise correlations between cells for the other datasets, as suggested by the reviewer. However, these other datasets use completely different cell types (including mouse embryonic stem cells), it is not clear to us that comparing correlations between these disparate datasets would provide information regarding the technical performance of the methods. Nevertheless, the high degree of concordant DNA methylation measurements between subsets of cells within each of our own datasets is consistent with a high technical reproducibility.

Strand-specific CpG methylation states at individual alleles in a single cell should be invariant. This provides a unique opportunity to assess the reproducibility of scDEEP-mC methylation calls by

calculating the methylation state concordance at individual CpGs in unique reads covering the same DNA strand from the same allele in the same cell. We calculated the proportion of identical methylation calls at all allele- and strand-resolved CpGs with coverage $\geq 2x$. We identified 34,700,605 allele- and strand-resolved CpG site measurements with coverage $\geq 2x$ in 145 mouse cells, and found that 0.27% of these were discordant, yielding a beta value that is not either 0 or 1. We identified 29,787,514 allele- and strand-resolved CpG site measurements with coverage $\geq 2x$ in 233 human cells, and found that 0.57% of these were discordant, yielding a beta value that is not either 0 or 1. However, we note that the vast majority of sites have a coverage of only $2x$, where even random methylation calling would yield a concordance of 50%. Thus, this raw fraction of identical calls is somewhat inflated. To correct for this, at each coverage depth, we subtracted the probability of measuring perfect agreement by random chance, then rescaled the result to [0, 1]. We then took the mean of this concordance across all coverage depths, weighted by the number of sites measured at each depth. This is analogous to the 'discordance' metric described in the Methods. Using these adjusted measures yielded a discordance of 0.46% for mouse cells and a discordance of 0.98% for human cells.

To investigate a potential representation bias introduced by G:C content, we measured the GC content and read depth in 50kb bins genome-wide. We added a density plot depicting the median read depth as a function of GC content for each method (including scDEEP-mC) in the supplemental information (Fig. S1).

2. Showing known maternal and paternal methylated sites in Figure 2d would enhance its interpretability.

Response:

We infer that this is in reference to Figure 3d (rather than Figure 2d), which illustrates a known imprinted locus. We have updated the figure by using colors and text to indicate which DMRs are maternally and paternally methylated.

3. Providing supplementary tables listing the identified AMRs and hemi-methylated regions would greatly benefit the community and enhance the value of the manuscript.

Response:

We agree that providing this data could be of assistance to the community and have uploaded allele-resolved methylation calls to the GEO database. We have additionally included a Supplementary Table S1 with the hemi-methylation source data for Figure 3e.

4. Visualization of the methylation status of the hemi-methylated region around the TFBS using a genome browser as an example would better illustrate the finding.

Response:

The data underlying the enrichment analyses reported in figure 3e is sparse for any single TFBS in any single cell, since only reads overlapping heterozygous SNPs with both strands represented on the same allele will yield informative data on hemimethylation. Thus, a genome browser track of a single site would not illustrate hemi-methylation in an informative manner. However, we have included a supplementary Figure with a meta-gene plot illustrating hemi-methylation in proximity to GATA3 and CDX2 binding sites, stratified by cell type, which helps to illustrate the distribution of hemi-methylation more clearly (Figure S5).

5. Many readers may be unfamiliar with solo-WCGWs. Therefore, a clear explanation of solo-WCGWs and the rationale for focusing on them is necessary.

Response:

We have added this sentence on page 12 to the last paragraph prior to the discussion to give further context on solo-WCGWs and explain why measuring their methylation level is relevant:

“Previous work has shown that isolated CpGs in a WCGW context, known as solo-WCGWs, tend to lose methylation in a predictable manner that is correlated with cumulative cell divisions.^{9,10}”

Reviewer #2 (Remarks to the Author):

Comments for the manuscript entitled “high-coverage allele-resolved single-cell DNA methylation profiling reveals cell lineage, X-inactivation state, and replication dynamics” by Spix et al.

The manuscript presents a significant technical advance in single-cell whole-genome bisulfite sequencing through the introduction of scDEEP-mC. The authors convincingly demonstrate that their method achieves high library complexity and CpG coverage, overcoming key limitations of previous approaches. Notably, the technique enables allele-resolved methylation analysis, detailed interrogation of hemi-methylation, and simultaneous measurement of replication dynamics. These capabilities open new avenues for investigating cell lineage, X-inactivation, and methylation maintenance. Overall, the manuscript offers a significant methodological advance with clear implications for single-cell epigenomics. I believe this work is suitable for publication in Nature Communications following clarification and streamlining of some methodological details, as outlined below:

1. Including additional quantitative comparisons—such as reproducibility metrics across independent replicates and performance across diverse cell types—would further strengthen the claims regarding the method’s scalability and general applicability.

Response:

As mentioned in our response to reviewer #1, reproducibility of DNA methylation measurements is difficult to quantify in single-cell studies, since each cell is unique and can only be measured once. We have computed pairwise methylation concordance between all cells in each species we analyzed (mouse and human) and report these in the supplemental figures (Fig. S3). We chose to measure concordance (fraction of sites with identical values) rather than correlation, as correlation can be destabilized by (mostly) binary data.

2. While the manuscript outlines the computational pipeline, additional details on how data sparsity is managed would benefit readers who are less familiar with single-cell bioinformatics.

Response: To clarify this topic, we have added the text below to the methods:

Data Sparsity:

Single-cell sequencing (and DNA methylation) data is often incomplete or sparse. Although scDEEP-mC facilitates high coverage compared to many existing methods, the resulting data is still sparse in nature, especially the allele-resolved methylation data (since allele assignment requires a heterozygous SNP near the CpG). The methods described below detail how this sparsity is handled in each analysis, but it may be helpful to describe some overarching principles which guided our analysis.

We do not make use of imputation, since this does not add any truly new information and may introduce noise (e.g. training data gathered from datasets with limited relevance). We also strive to avoid summarizing methylation values over large genomic bins that are defined without reference to underlying biology (e.g., mean beta over 100kb bins). Rather, we reason that a few high-quality, informative sites with high biological significance are preferable to a large number of noisy or uninformative sites. Thus, we strive to restrict our analyses to exactly the CpGs of interest. These methylation calls are then often summarized per-region or per-cell, discarding regions or cells with

insufficient data. In this way, we allow for missing data and non-overlapping coverage between cells without obscuring underlying biological differences. For dimension reduction, we prefer non-negative matrix factorization (NMF), as implemented by the RcppML package, since it natively accommodates missing values through sparse matrices.

3. Figures 2a and 2g are challenging to interpret. Enhancing the figure legends or adding more detailed descriptions in the main text would improve reader understanding of these key data presentations.

Response: We have updated both the text and figure legends to clarify these panels. Revisions are included below:

2a body:

For each cell, we measured the mean beta for all CpGs in each of 50,286 cell-type-specifically hypomethylated regions cataloged by Loyfer and colleagues²⁰, then summarized by target cell type. In this analysis, cells corresponding closely to the target cell type will display significant hypomethylation at the relevant regions.

2a legend:

Columns represent individual cells, while rows represent aggregated cell-type-specific region sets. Gastrointestinal epithelial and lymphocytic cell types can be clearly distinguished, as well as several doublets incorporating different cell types with intermediate beta values.

2g body:

We applied rank-2 NMF to raw beta values of the 75% most variable CpGs across all cells, essentially decomposing this high-dimensional methylation data into two dimensions. This analysis revealed two major cell clusters (lymphocytes and intestinal epithelial cells), recapitulating the cell type groupings previously described, and partitioning doublets and putative G2-phase cells (Fig. 2f). Overlaying the mean beta value in regions specifically hypomethylated in intestinal epithelial cells or T cells further confirmed the accuracy of cell type inference by NMF (Fig. 2g).

2g legend:

(g) Cell-type-specific hypomethylation (as in panel [a]) and solo-WCGW methylation values overlaid on each cell in the NMF map.

4. I did not find an explanation for Figure 4b in the main text. Please ensure that all figures are adequately described and integrated into the manuscript narrative.

Response: We added a note to the text that explains Figure 4b.

5. A discussion of the limitations of scDEEP-mC compared with other published methods would be valuable. Specifically, addressing any potential drawbacks or trade-offs inherent to the approach will help readers contextualize the advance.

Response:

We have added a section to the manuscript discussing these considerations:

"The plate-based nature of scDEEP-mC limits the number of cells that can be analyzed in parallel, although miniaturization of the protocol has the potential to increase cellular throughput in the future. The deep sequencing performed on each cell also reduces the number of cells that can be analyzed for a given number of sequence reads. As with other random priming-based methods, scDEEP-mC is subject to some random priming bias (Fig. S2), although this effect has been partially mitigated by thoughtful design of the random primers."

We also added a supplementary figure depicting the observed and expected coverage across different genomic regions (CpG islands, gene bodies, intergenic regions, etc) for several methods, which helps to quantify the extent of this random priming bias (Fig. S2).

Reviewer #2 (in attachment): Methodological Detail. The manuscript would benefit from more detailed descriptions of key methodological steps, such as the optimization of primer concentrations and the strand-specific library construction. Providing these details will improve clarity without sacrificing necessary technical rigor.

Response:

We have added a section to the methods to address these considerations:

We performed several experiments to determine the optimal concentration of first-strand random primer and found that using too much random primer resulted in excessive adapter contamination upon sequencing, while using too little random primer decreased the probability of successfully preparing a library (data not shown). We found that the concentration reported below to be a good balance between these challenges for euploid human and mouse cells, with minimal adapter contamination (Fig. 1c, f) and high success rates. If scDEEP-mC is applied to other inputs, such as haploid gametes or small cell pools, it may be necessary to empirically titrate the primer concentration for these samples. (Notably, we found the protocol to be less sensitive to the concentration of second-strand random primer.)

Reviewer #3 (Remarks to the Author):

This manuscript presents scDEEP-mC, a promising technology for generating high-coverage single-cell DNA methylation sequencing data. The authors demonstrate its utility by conducting allele-resolved methylation analysis and validating key epigenetic phenomena. However, several concerns require further clarification:

(1). In Figure 1, The comparison of scDEEP-mC to other methods appears to be confounded by significant differences in total number of reads per cell. Please provide a summary table or figure showing the median number of reads per cell for each dataset before downsampling. This will allow for a more accurate assessment of method performance and avoid potential biases introduced by sequence depth.

For example, If scDEEP-mC has a significantly higher total number of reads (> 20M per cell), while other method has <2M reads/cell, then, the improved performance (genome coverage) could be attributed to the increased sequencing depth rather than inherent methodological advantages.

Response:

*We appreciate the reviewer's thoughtful critique. Coverage is indeed related to both sequencing depth and library complexity. However, some methods (which produce high-complexity libraries) cannot be sequenced deeper because of limiting library **quantities**. On the other hand, other methods produce larger library quantities and can be sequenced very deeply. Figures 1c and 1f focus on library quality by downsampling to a fixed level that offers a fair comparison among methods regardless of sequencing depth. Figures 1d and 1g explore higher read counts to accommodate methods that can be sequenced deeper.*

(2). What's the minimal coverage required for accurate allele-resolved methylation analyses and X-

inactivation studies in single cells. This information would provide valuable guidance for future studies and experimental design.

Response:

Two factors interact to determine the density of allele-resolved methylation data – the density of heterozygous SNPs identified, and the coverage surrounding those SNPs. We targeted a depth of 30-35M 150bp paired-end reads per cell, as scDEEP-mC libraries typically are close to saturation at this depth. (We added a sentence to the methods explaining this.) It is not possible to identify a 'minimal coverage' necessary for X-inactivation or allele-resolved methylation studies, as this depends on the goals and requirements of the particular study, but we suggest that experimenters making use of our method plan to acquire 35M reads per cell.

(3). mCH methylation levels vary across tissues (e.g., high (~ 6%) in neurons, low in other cell types). However, the impact of this variation on scDEEP-mC performance is not adequately addressed. Given that the initial primer design assumed negligible mCH methylation (only 1% C in CpG context during the design of random primer), its applicability to tissues with high mCH levels, such as brain, may be limited. The authors should discuss the potential implications of this assumption on the accuracy and robustness of scDEEP-mC in different tissues, especially the mCH in brain.

Response:

Although the random primer sequence does not include CpH, the bulk of the sequence data acquired from scDEEP-mC does not originate from the random priming sequence, but rather from adjacent sequence. However, in cells with extensive CpH methylation, this might introduce some bias. We have added a note to the discussion section addressing this:

"Finally, we note that the design of the random primers does not account for methylation of cytosines in CpH context. Although CpH methylation is rare in most cell types, it may be found at non-negligible levels in e.g. neuronal cells.⁸ In this setting, our primer design could bias against regions with high CpH methylation, although CpH methylation is still accurately reported by scDEEP-mC, since only the first nine bases of each read is generated via random priming."

(4). The manuscript lacks a comprehensive assessment of the performance of the DNA methylation phasing pipeline. Metrics such as accuracy, precision, and recall should be evaluated and presented to demonstrate the reliability of this pipeline.

Response:

There are two facets of the allele-resolved methylation analysis that contribute to final accuracy – single-nucleotide variant discovery, and phasing accuracy.

We tested the accuracy of our variant calling pipeline in mouse by comparing to the gold-standard variant calls made available by the Mouse Genome Project and found a precision of 93.5% and recall of 67.4%. We also performed whole-genome sequencing of 4 additional mice of the same litter (to account for strain- or source-specific variation) and found a precision of 90.8% and recall of 69.5% compared to this dataset.

To evaluate the accuracy of our methylation phasing pipeline, we generated simulated bisulfite-converted reads from two synthetic mouse genomes representing two parental alleles (using the Sherman tool). We then processed these reads using our allele-resolved methylation pipeline and measured the number of correctly assigned, incorrectly assigned, and unassigned reads. We found that our algorithm had a precision of 99.97% and recall of 76.76%.

We have added additional text and figures to the supplementary material detailing the methods and results of these analyses (Fig. S4).

(5). The description of the methods used for Figure 4 is unclear. It is stated that data was downloaded

from reference 27 (line 343) in the method part, but the analysis seems to involve data generated using scDEEP-mC (line 243) in the result part. Please clarify which analyses were performed on the downloaded dataset and which utilized data generated by scDEEP-mC.

Response:

We have updated the text and methods to clarify this analysis. Publicly available data was used to define replication timing across the genome. We then summarized our scDEEP-mC data by replication timing.

Minor:

(1). line 247, "higher rates of hemi-methylation in early passage cells", should be "higher rates of hemi-methylation than early passage cells"?

Response:

We thank the reviewer for pointing out this potentially ambiguous phrasing. We changed this to read:

"In early passage cells, late-replicating loci have higher proportions of hemi-methylation"

which we hope is clearer.

Reviewer #3 (Remarks on code availability):

The link to the code is not accessible.

Response:

*We have uploaded the code to **Code Ocean**, which should be available to the reviewer prior to publication. We have also deposited the code in a Zenodo repository, which can be accessed using the following link: <https://doi.org/10.5281/zenodo.15350891>*

The Data Access section now reads:

Data Access:

Code:

Code used in these analyses is available in a Zenodo repository, which can be accessed using the following link: <https://doi.org/10.5281/zenodo.15350891>

Data:

scDEEP-mC data were deposited into the Gene Expression Omnibus database under accession number GSE280161 and are available at the following URL:

<https://www.ncbi.nlm.nih.gov/geo/query/acc.cgi?acc=GSE280161>

Currently a token is needed for access. Access will be made public prior to publication.

Access Token: *abcvieimnfarxqz*

ROUND 1 REVIEWER 1 ATTACHMENT:

Comments for the manuscript entitled “high-coverage allele-resolved single-cell DNA methylation profiling reveals cell lineage, X-inactivation state, and replication dynamics” by Spix et al.

The manuscript presents a significant technical advance in single-cell whole-genome bisulfite sequencing through the introduction of scDEEP-mC. The authors convincingly demonstrate that their method achieves high library complexity and CpG coverage, overcoming key limitations of previous approaches. Notably, the technique enables allele-resolved methylation analysis, detailed interrogation of hemi-methylation, and simultaneous measurement of replication dynamics. These capabilities open new avenues for investigating cell lineage, X-inactivation, and methylation maintenance. Overall, the manuscript offers a significant methodological advance with clear implications for single-cell epigenomics. I believe this work is suitable for publication in Nature Communications following clarification and streamlining of some methodological details, as outlined below:

1. **Methodological Detail.** The manuscript would benefit from more detailed descriptions of key methodological steps, such as the optimization of primer concentrations and the strand-specific library construction. Providing these details will improve clarity without sacrificing necessary technical rigor.
2. **Quantitative Comparisons.** Including additional quantitative comparisons—such as reproducibility metrics across independent replicates and performance across diverse cell types—would further strengthen the claims regarding the method’s scalability and general applicability.
3. **Handling Data Sparsity.** While the manuscript outlines the computational pipeline, additional details on how data sparsity is managed would benefit readers who are less familiar with single-cell bioinformatics.
4. **Figure Clarity.** Figures 2a and 2g are challenging to interpret. Enhancing the figure legends or adding more detailed descriptions in the main text would improve reader understanding of these key data presentations.
5. **Explanation of Figure 4b.** I did not find an explanation for Figure 4b in the main text. Please ensure that all figures are adequately described and integrated into the manuscript narrative.
6. **Comparison with Existing Methods.** A discussion of the limitations of scDEEP-mC compared with other published methods would be valuable. Specifically, addressing any potential drawbacks or trade-offs inherent to the approach will help readers contextualize the advance.